# Molecular basis for catabolism of the abundant metabolite *trans*-4-hydroxy-L-proline by a microbial glycyl radical enzyme

Lindsey RF Backman[1†], Yolanda Y Huang[2†‡], Mary C Andorfer[3], Brian Gold[1§], Ronald T Raines[1], Emily P Balskus[2]*, Catherine L Drennan[1,3,4]*

[1]Department of Chemistry, Massachusetts Institute of Technology, Cambridge, United States; [2]Department of Chemistry and Chemical Biology, Harvard University, Cambridge, United States; [3]Department of Biology, Massachusetts Institute of Technology, Cambridge, United States; [4]Howard Hughes Medical Institute, Massachusetts Institute of Technology, Cambridge, United States

*For correspondence:
balskus@chemistry.harvard.edu
(EPB);
cdrennan@mit.edu (CLD)

[†]These authors contributed equally to this work

Present address: [‡]Department of Environmental Genomics and Systems Biology, Lawrence Berkeley National Laboratory, Berkeley, United States; [§]Department of Chemistry & Chemical Biology, University of New Mexico, Albuquerque, United States

**Competing interests:** The authors declare that no competing interests exist.

**Abstract** The glycyl radical enzyme (GRE) superfamily utilizes a glycyl radical cofactor to catalyze difficult chemical reactions in a variety of anaerobic microbial metabolic pathways. Recently, a GRE, *trans*-4-hydroxy-L-proline (Hyp) dehydratase (HypD), was discovered that catalyzes the dehydration of Hyp to (*S*)-$\Delta^1$-pyrroline-5-carboxylic acid (P5C). This enzyme is abundant in the human gut microbiome and also present in prominent bacterial pathogens. However, we lack an understanding of how HypD performs its unusual chemistry. Here, we have solved the crystal structure of HypD from the pathogen *Clostridioides difficile* with Hyp bound in the active site. Biochemical studies have led to the identification of key catalytic residues and have provided insight into the radical mechanism of Hyp dehydration.

## Introduction

The microbes that inhabit the human body, collectively referred to as the human microbiome, catalyze a diverse range of chemical reactions that can have profound impacts on human health (*Sharon et al., 2014*; *Joice et al., 2014*; *Koppel et al., 2017*). The most densely populated microbial environment among human body sites is the gastrointestinal (GI) tract with an estimated $10^{11}$ bacterial cells per gram (*Tropini et al., 2017*; *Sender et al., 2016*). Due to the largely anoxic nature of the GI tract, this body site is inhabited by facultative and obligate anaerobes that perform a wide variety of challenging chemical transformations. Recent studies suggest correlations between changes in gut microbiome composition and diseases, such as metabolic disorders, cardiovascular diseases, autoimmune diseases, and neurological disorders (*Sharon et al., 2014*; *Kang et al., 2013*). Although it is clear that the gut microbiome plays a significant role in human health, the biochemical reactions governing bacterial-host homeostasis remain unclear.

Elucidating the activities of uncharacterized enzymes present in the human gut microbiome can enhance our understanding of this microbial community. *trans*-4-Hydroxy-L-proline (Hyp) dehydratase (HypD) is a newly discovered glycyl radical enzyme (GRE) that catalyzes the transformation of Hyp to (*S*)-$\Delta^1$-pyrroline-5-carboxylate (P5C) and water (*Figure 1A*; *Levin et al., 2017*). Hyp is an abundant nutrient in the GI tract. Generated from hydroxylation of Pro residues within proteins like collagen (*Gorres and Raines, 2010*), this nonproteinogenic amino acid represents the most abundant post-translational modification in mammals and is also found in the human diet (*Vázquez-Ortíz et al., 2004*; *Verbeken et al., 2003*; *Valiente et al., 1995*). Bioinformatic analyses

**Figure 1.** Hyp dehydration is catalyzed by the GRE HypD in a prominent gut microbial metabolic pathway. (A) Anaerobic microbial metabolism of *trans*-4-hydroxy-L-proline (Hyp) is catalyzed by Hyp dehydratase (HypD), a glycyl radical enzyme (GRE). The product of this transformation, (S)-$\Delta^1$-pyrroline-5-carboxylate (P5C), is an intermediate in many primary metabolic pathways. Hyp can be used to generate ATP for energy metabolism, converted to other amino acids for protein synthesis, and catabolized to form sources of carbon and nitrogen. (B) General mechanism proposed for GREs. A [4Fe-4S]-cluster dependent radical *S*-adenosylmethionine (AdoMet) activating enzyme (AE) generates a radical species on a conserved glycine residue in the GRE, using *S*-adenosylmethionine and an electron, and forming 5'-deoxyadenosine (5'dA) and methionine. The glycyl radical generates a thiyl radical species on a conserved Cys, and this thiyl radical initiates catalysis by abstracting a hydrogen atom from the substrate (S). Upon product (P) formation, the thiyl radical is regenerated to complete the catalytic cycle.

The online version of this article includes the following figure supplement(s) for figure 1:

**Figure supplement 1.** Transformations catalyzed by GRE eliminases.

**Figure supplement 2.** Enzymatic transformations involving 5-membered heterocyclic substrates.

---

of metagenomes indicate HypD is one of the most abundant GREs in the healthy human gut microbiome, thus suggesting it plays an important role in many bacteria (*Levin et al., 2017*). P5C is a central metabolite in amino acid biosynthetic pathways, providing a source of building blocks for protein synthesis and/or of carbon and nitrogen (*Figure 1A*). In gut Clostridiales, Hyp enables amino acid fermentation, also known as Stickland fermentation (*Stickland, 1934*). These organisms encode HypD near a P5C reductase (P5CR), which reduces the product P5C to L-proline (Pro), which can undergo further reduction. The formation of Pro as a downstream metabolite during Hyp fermentation by Clostridiales results in the chemical reversal of the post-translational modification of Pro to Hyp, which was not previously thought to occur. In addition to commensal Clostridiales, HypD is found in the prominent antibiotic-resistant opportunistic pathogen *Clostridioides difficile*, formerly known as *Clostridium difficile*. In 2015, *C. difficile* was responsible for approximately 500,000 infections and 29,000 deaths, making this pathogen a major health concern (*Leffler and Lamont, 2015*; *Lessa et al., 2015*). As a key metabolic enzyme, with no protein homolog in humans, HypD could be a promising antibiotic target for *C. difficile* and other pathogens.

The discovery of HypD also revealed a previously unknown enzymatic activity, expanding the known chemistry of the GRE superfamily. This evolutionarily ancient protein superfamily is essential for anaerobic primary and secondary metabolism in bacteria and archaea. All GREs use a glycine-centered radical cofactor for catalysis. Briefly, these enzymes are activated by a cognate *S*-adenosylmethionine (AdoMet)-dependent activating enzyme (AE), generating a protein-based radical on a conserved Gly residue positioned within the GRE active site (*Figure 1B*; *Conradt et al., 1984*). The glycyl radical in turn generates a catalytically essential thiyl radical on a conserved Cys, which

initiates the chemical transformation. Upon product formation, the thiyl radical is regenerated and the radical species is returned to the Gly residue for storage.

HypD is part of the eliminase class of GREs, which includes glycerol dehydratase (GD), propanediol dehydratase (PD), choline trimethylamine-lyase (CutC), and the recently identified isethionate sulfite-lyase (IslA) (*Backman et al., 2017*; *O'Brien et al., 2004*; *LaMattina et al., 2016*; *Kalnins et al., 2015*; *Xing et al., 2019*). Interestingly, several of the eliminases catalyze transformations with reactivities that are also performed by adenosylcobalamin (coenzyme $B_{12}$)-dependent enzymes. For example, dehydration of 1,2-propanediol and glycerol was first discovered in the 1960s as an activity catalyzed by a $B_{12}$-dependent propanediol dehydratase (*Zagalak et al., 1966*; *Rétey et al., 1966*). Additionally, choline cleavage by CutC resembles the activity of ethanolamine ammonia-lyase, a $B_{12}$-dependent enzyme that catalyzes an analogous C–N bond cleavage of ethanolamine using radical chemistry (*Craciun and Balskus, 2012*; *Mori et al., 2014*). In contrast, dehydration of Hyp to P5C does not have precedence in $B_{12}$ enzymology, nor does the C–S bond cleavage of IslA (*Xing et al., 2019*). In particular, the oxidation of a C–N bond that occurs in HypD is unprecedented among GREs since all other eliminases catalyze the oxidation of a C–O bond to generate aldehyde products (*Figure 1—figure supplement 1*).

Another notable feature of HypD is that its substrate is a conformationally constrained pyrrolidine ring lacking a freely rotatable $C^\alpha$–$C^\beta$ bond (*Figure 1—figure supplements 1–2*). Among GREs, the enzyme with the most similar substrate is the class III ribonucleotide reductase (RNR), which also acts on a substrate with a 5-membered heterocyclic ring (*Figure 1—figure supplement 2*). The reaction of RNR, however, is more similar to that of diol dehydratases than it is to HypD. Perhaps the enzyme-catalyzed reaction most similar to the reaction catalyzed by HypD is the dehydration of the ribose-containing substrate cytidine triphosphate (CTP) performed by the recently characterized AdoMet radical enzyme viperin (*Gizzi et al., 2018*). This reaction is reminiscent of Hyp dehydration because it involves the elimination of a hydroxyl group on the carbon position β to a C–O bond within the 5-membered ring (*Gizzi et al., 2018*), although it lacks the C–N oxidation catalyzed by HypD (*Figure 1—figure supplement 2*).

In order to gain insight into the chemical mechanism of HypD, we set out to elucidate the structure of this GRE to identify active site residues that may be important for catalysis. This information could guide analysis of microbiome sequencing data and development of enzyme inhibitors. Here, we present a Hyp-bound structure of HypD from *C. difficile* 70-100-2010 along with biochemical assays performed with enzyme variants and deuterated substrate to better understand how this newly discovered GRE eliminase performs Hyp dehydration.

## Results

### Overall architecture of HypD is similar to other GRE eliminases

A structure of HypD from *C. difficile* 70-100-2010 was solved by molecular replacement to 2.05 Å resolution using the GRE homolog CutC (PDB: 5FAU) (*Bodea et al., 2016*) as the search model (*Table 1*). During model refinement, we observed electron density resembling glycerol in the active site (*Figure 2—figure supplement 1*). Although we believe that glycerol binding is an artifact (glycerol was used as a cryoprotectant), it is not surprising that glycerol is able to bind given HypD's high sequence similarity to GD. To obtain a Hyp-bound structure, we used a different cryoprotectant (see Methods) and included Hyp in both the crystallization buffer and cryoprotectant solution. This second HypD structure was solved to 2.52 Å resolution by molecular replacement using the glycerol-bound HypD structure as the search model (*Table 1*).

In agreement with all characterized GREs (*O'Brien et al., 2004*; *LaMattina et al., 2016*; *Kalnins et al., 2015*; *Xing et al., 2019*; *Bodea et al., 2016*), HypD is dimeric with each monomer consisting of two five-stranded half β-barrels, anti-parallel to one another and surrounded by α-helices (*Figure 2A*). The active site is buried within the center of the barrel, which is thought to protect radical species from being quenched by solvent. Two loops essential for catalysis are juxtaposed in the active site: the Gly loop and the Cys loop (*Figure 2A*). The Gly loop contains the conserved Gly residue (Gly765 in HypD) and is part of the C-terminal glycyl radical domain that is found in all GREs (*Backman et al., 2017*) whereas the neighboring Cys loop contains the catalytic Cys434 (*Figure 2A–*

**Table 1.** Data collection and model refinement statistics for crystallography.
Values in parentheses denote highest resolution bin.

| | HypD with glycerol bound | HypD with Hyp bound |
|---|---|---|
| Space group | P2$_1$ | P2$_1$ |
| Unit cell (Å) | 100.3, 341.7, 122.6, 90.0°, 107.1°, 90.0° | 101.2, 350.2, 124.5, 90.0°, 105.7°, 90.0° |
| Resolution (Å) | 50–2.05 (2.09–2.05) | 50–2.52 (2.59–2.52) |
| R$_{sym}$ | 16.8 (75.7) | 20.4 (97.5) |
| CC$_{1/2}$ | 99.0 (58.8) | 99.4 (72.1) |
| <I/σ> | 8.40 (1.82) | 10.75 (2.12) |
| Completeness (%) | 99.0 (98.3) | 99.7 (99.4) |
| Unique reflections | 486251 (24062) | 278476 (44812) |
| Total reflections | 1626596 (168674) | 1944676 (294711) |
| Redundancy | 7.07 (7.01) | 6.98 (6.58) |
| R$_{work}$/R$_{free}$ | 0.166/0.193 | 0.186/0.224 |
| RMSD bond length (Å) | 0.007 | 0.008 |
| RMSD bond angles (°) | 0.86 | 0.966 |
| Chains in asymmetric unit | 8 | 8 |
| Number of: | | |
| Total atoms | 54954 | 52103 |
| Protein atoms | 49994 | 49851 |
| Water molecules | 4834 | 2180 |
| Gol/Hyp | 48 | 72 |
| Ramachandran analysis | | |
| Favored (%) | 98.16 | 97.71 |
| Allowed (%) | 1.71 | 1.99 |
| Disallowed (%) | 0.13 | 0.30 |
| Rotamer outliers (%) | 1.46 | 3.27 |
| Average B factors | | |
| Protein (Å$^2$) | 21.0 | 35.8 |
| Water (Å$^2$) | 27.2 | 27.7 |
| Gol/Hyp (Å$^2$) | 22.3 | 31.2 |

**C**). Gly765 is located 3.9 Å from Cys434 (**Figure 2C**), consistent with radical transfer from Gly765 to Cys434 to generate a transient thiyl radical for initiation of catalysis on substrate.

## Positioning of Hyp in active site suggests hydrogen atom abstraction from C5 of Hyp

In the active site of the Hyp-bound HypD structure, electron density was observed for the substrate Hyp (**Figure 2B**). To investigate whether Hyp binds to HypD in the C$^\gamma$-*exo* or C$^\gamma$-*endo* pucker state (**Figure 2—figure supplement 2**), we modeled and refined both configurations of Hyp into the electron density and found that the electron density maps are best fit by a C$^\gamma$-*exo* configuration (**Figure 2—figure supplement 3**). Additionally, we calculated the energies associated with these two configurations of zwitterionic Hyp (see Methods) and found the C$^\gamma$-*exo* pucker to be more energetically favorable than the C$^\gamma$-*endo* pucker by 2.7 kcal mol$^{-1}$. A preference for the C$^\gamma$-*exo* pucker has also been noted for free Hyp (**Lesarri et al., 2005**) and peptidyl Hyp in the context of collagen structure (**Shoulders and Raines, 2009**). The C$^\gamma$-*exo* puckering of the pyrrolidine ring orients C5 of Hyp in closest proximity to Cys434, prompting us to propose C5 as the site of hydrogen atom abstraction. The distance between C5 and the sulfur atom of Cys434 is 3.8 Å (**Figure 2C**), a typical distance

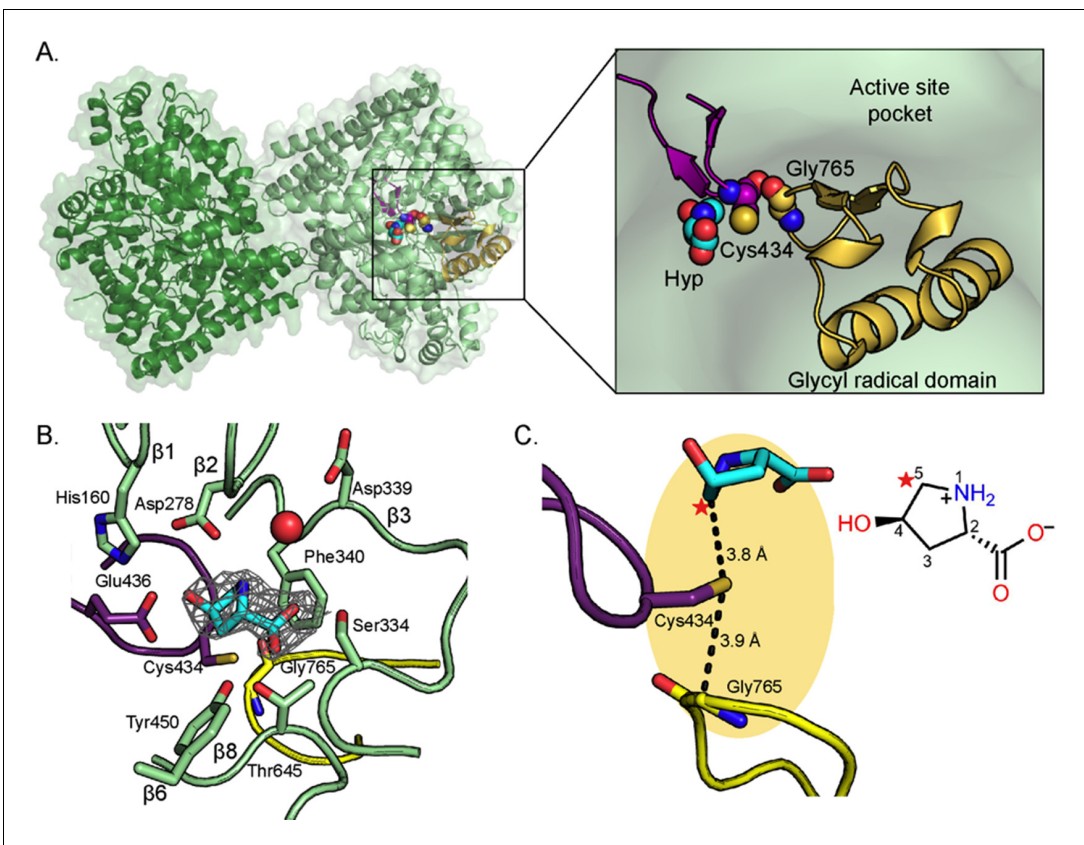

**Figure 2.** Structure of *C. difficile* HypD with Hyp bound. (**A**) Dimeric structure of HypD (green) with the glycyl radical domain that houses the Gly loop in yellow and the Cys loop in purple. Gly765, Cys434, and Hyp are shown in spheres. (**B**) $2F_o$-$F_c$ maps (contoured at $1.0\sigma$, gray) indicate electron density for Hyp. Hyp is positioned above the Gly loop (yellow) and Cys loop (purple) with residues from the strands of the $\beta$-barrel (green) forming the sides of the active site. A water molecule is shown as red sphere. (**C**) C5 of HypD (starred) is the closest atom to the catalytic Cys (Cys434), which is found in the active site within van der Waals distance of Gly765, the site of the glycyl radical. See *Figure 8A* for additional distances between Hyp and Cys434.

The online version of this article includes the following source data and figure supplement(s) for figure 2:

**Source data 1.** Cartesian coordinates for zwitterionic Hyp in $C^\gamma$-*exo* pucker calculated from DFT.
**Source data 2.** Cartesian coordinates for zwitterionic Hyp in $C^\gamma$-*endo* pucker calculated from DFT.
**Figure supplement 1.** A 2.05 Å resolution structure of HypD with glycerol bound in the active site.
**Figure supplement 2.** HypD conformers generated by DFT calculations.
**Figure supplement 3.** Comparison of electron density maps for $C^\gamma$-*exo* Hyp versus $C^\gamma$-*endo* Hyp modeled into HypD active site.

between the site of hydrogen atom abstraction and the catalytic cysteine in other structurally characterized GREs (*Backman et al., 2017*; *Vey et al., 2008*). We calculate that in the observed $C^\gamma$-*exo* pucker conformation, the dihedral angle between the amino group and the hydroxyl leaving group of Hyp is 75.8°, similar to the dihedral angles (~60°) observed for substrates bound to other GRE eliminases (*Figure 3—figure supplement 1*; *O'Brien et al., 2004*; *LaMattina et al., 2016*; *Bodea et al., 2016*). In contrast, Hyp in the $C^\gamma$-*endo* pucker conformation has a dihedral angle of 165° between the amino group and the hydroxyl leaving group. Hyp is distinct from other GRE eliminase substrates in that it contains a constrained 5-membered ring and thus has decreased conformational flexibility; the similarity in the conformations of different GRE eliminase substrates may indicate an important role for this binding mode in catalysis.

## Hyp leaving group is pointing toward a conserved CXE motif

Hyp is positioned in the active site through a series of hydrogen bonds that include: Glu436 of the Cys loop, Asp278 of the β1 strand, Ser334 of a short α-helix between the β2 and β3 strands, Thr645 of β8 strand, and a water molecule that interacts with Asp339 of the β3 strand (*Figure 2B*). In addition, hydrophobic contacts may play a role in the positioning of Hyp (*Figure 3D*). The Cys loop of HypD contains the CXE motif (Cys434-Val435-Glu436) that is conserved among GRE eliminases (*Figure 3A–C*; *Figure 3—figure supplement 2A–C*; *O'Brien et al., 2004*; *LaMattina et al., 2016*; *Xing et al., 2019*; *Bodea et al., 2016*). In previous GRE eliminase structures, the Glu of the CXE motif is observed to interact with a hydroxyl group of substrate through a hydrogen bond (*Figure 3B–C*; *Figure 3—figure supplement 2B–C*; *O'Brien et al., 2004*; *LaMattina et al., 2016*; *Xing et al., 2019*; *Bodea et al., 2016*). In these cases, the Glu is proposed to deprotonate the substrate's hydroxyl group to form a ketyl radical intermediate that aids in the elimination of a different hydroxyl group (diol dehydratases), trimethylamine (CutC), or sulfite (IslA) (*Figure 3*; *Figure 1—figure supplement 1*; *Figure 3—figure supplement 3*; *O'Brien et al., 2004*; *LaMattina et al., 2016*; *Xing et al., 2019*; *Bodea et al., 2016*). Thus in these GREs, the hydroxyl group, which is ultimately converted to a product aldehyde, is pointing toward and interacting with the Glu of the CXE motif, and the substrate's leaving group is pointing away from the Glu. In HypD, we also find that Glu436 is positioned to hydrogen bond with the hydroxyl group of Hyp (*Figure 3A*). However, in HypD, the Hyp hydroxyl group <u>is</u> the leaving group, and therefore it is the leaving group that points toward and interacts with the Glu of the CXE motif rather than pointing away as is the case for the other

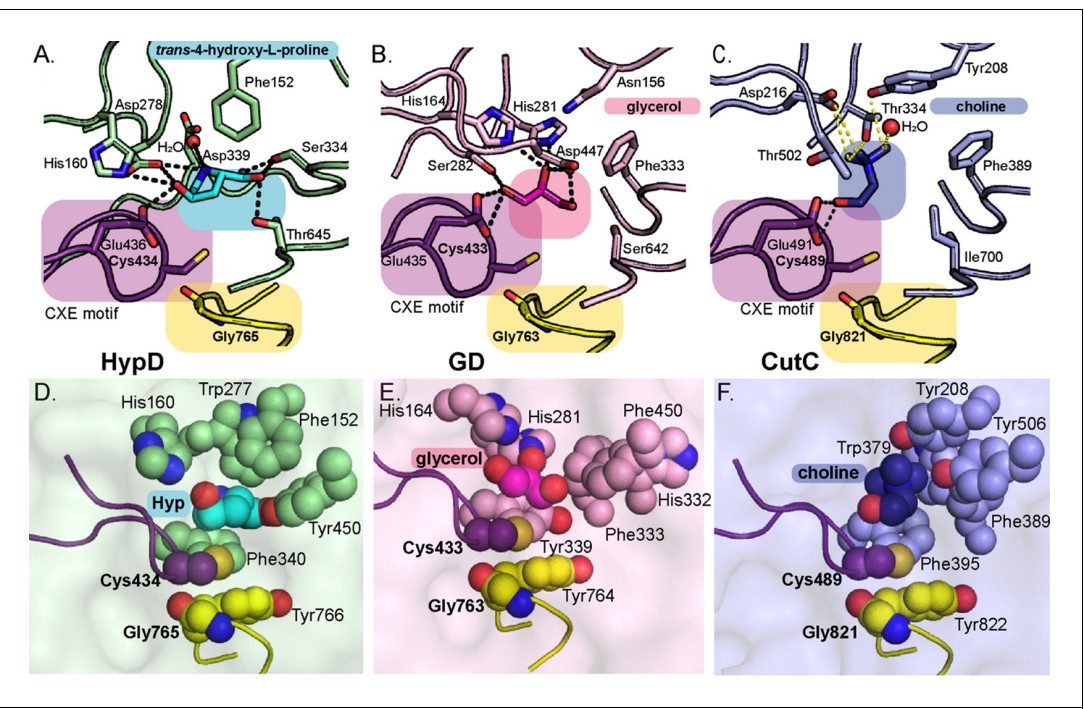

**Figure 3.** Active site of HypD has unique features that enable Hyp dehydration. Conserved Gly and Cys loops in addition to active site residues are displayed for (**A**) HypD, (**B**) GD, and (**C**) CutC. PDB-deposited structures for GD (1R9D) and CutC (5FAU) were used to generate this figure. The Cys loop is highlighted in purple and the Gly loop is highlighted in yellow. Hydrogen bonds are shown as black dashed lines and CH–O interactions as yellow dashed lines. Aromatic residues in the active sites of HypD (**D**), GD (**E**), and CutC (**F**) aid in substrate packing. The online version of this article includes the following figure supplement(s) for figure 3:

**Figure supplement 1.** Dihedral angles of substrates bound in GRE eliminases.

**Figure supplement 2.** Active sites of propanediol dehydratase (PD) and isethionate sulfite-lyase (IslA) compared to HypD.

**Figure supplement 3.** Proposed mechanism for CutC elimination reaction.

**Figure supplement 4.** A multiple sequence alignment of putative HypDs with characterized GREs.

GRE eliminases. Thus, the orientation of the substrate's leaving group with respect to the CXE is reversed and the fate of this hydroxyl group is unique in the HypD-catalyzed reaction.

Outside of the Cys and Gly loop motifs, there are no strictly conserved sequences among GRE eliminases (*Figure 3—figure supplement 4*). Each eliminase is adapted to accommodate its substrate and HypD is no exception. In addition to interacting with Glu436, the hydroxyl group of Hyp is also within hydrogen bonding distance of Asp278 (*Figure 4A*). The amino group of Hyp, which must be deprotonated as part of the reaction cycle, makes interactions with Asp278 and Asp339, the latter through water (*Figure 4A*). Notably, these water interactions are present in all active sites of the 8 molecules in the asymmetric unit. Asp339 and Asp278 are strictly conserved in HypD but are not conserved in other GRE eliminases (*Figure 3—figure supplement 4*).

Interestingly, HypD uses three polar, uncharged residues (Ser334, Thr645, Tyr450) to hydrogen bond to the Hyp carboxylate moiety (*Figure 4B*), all of which are conserved in HypD (*Figure 3—figure supplement 4*). Ser334 is positioned in the space that is often occupied by a Phe residue in other GRE eliminases (Phe333 in GD, Phe389 in CutC, Phe338 in PD, and Phe680 in IslA) (*Figure 3A–C*; *Figure 3—figure supplement 2A–C*). Arg156 and Lys326 may contribute to charge stabilization of the Hyp carboxylate, although they do not directly interact with substrate (*Figure 4D*).

As is the case for other GREs, HypD has many aromatic residues in its active site (*Figure 3D–F*; *Figure 3—figure supplement 2D–F*; *Backman et al., 2017*). Possible roles of these residues include

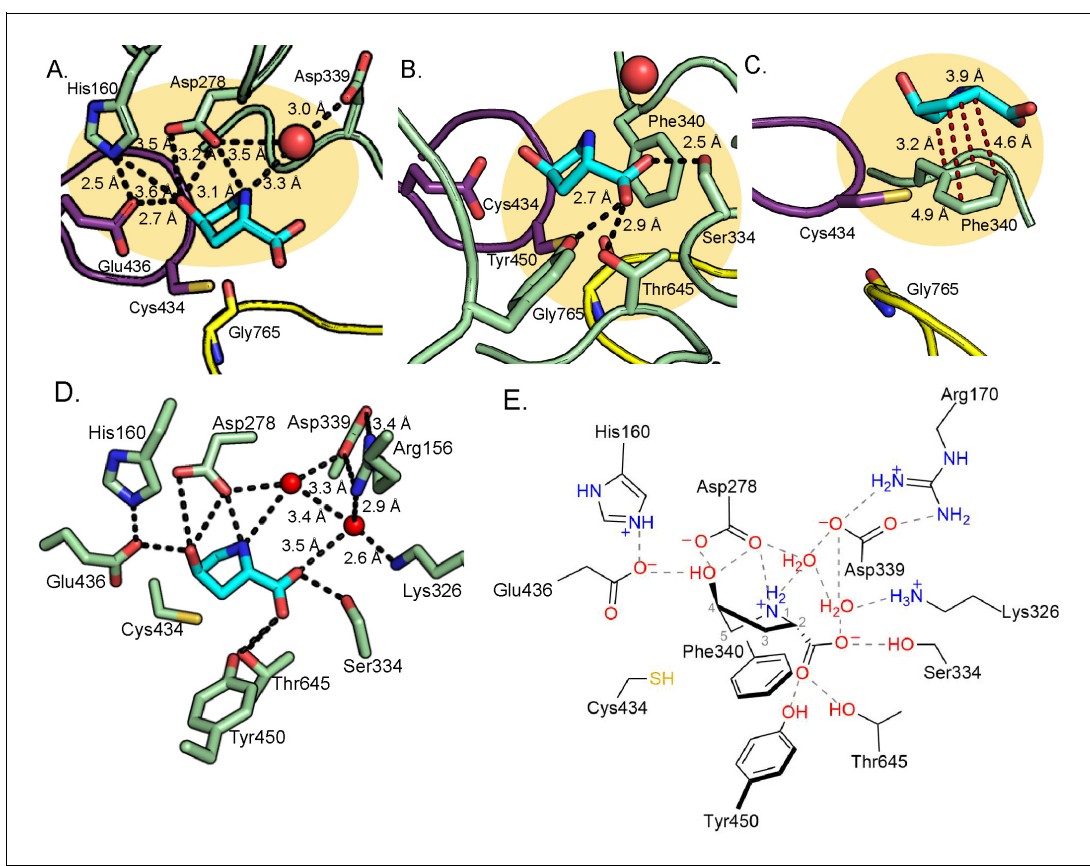

**Figure 4.** Hydrogen bonding and proline-aromatic interactions with nearby residues and bound water molecules allow for HypD chemistry. (**A**) Residues and ordered water molecule (red sphere) that are within hydrogen bonding distance to the hydroxyl and amine of Hyp shown with corresponding distances. (**B**) Residues within hydrogen bonding distance of the carboxylate group of Hyp. (**C**) Phe340 is positioned such that it could make proline-aromatic interactions (shown as red dashed line) with Hyp. (**D**) An extended hydrogen bonding network among Hyp, protein residues, and water molecules (red spheres) is observed in the active site. (**E**) Diagram of protein and water interactions with Hyp. All hydrogen bonds are indicated with gray dashed lines. Distances (Å) can be found in panels **A-D**.

exclusion of solvent, creation of hydrophobic packing interactions with substrates, and/or stabilization of substrate and reaction intermediates through cation-π interactions. In HypD, Phe152, Trp277, and Phe340 are in contact with the substrate, with Phe340 positioned to stabilize Hyp through a proline-aromatic interaction, which is characteristically similar to a cation-π interaction (*Figure 4C*). Previous studies have shown that energetically favorable interactions occur between proline and aromatic residues within proteins (*Zondlo, 2013*). These favorable interactions are thought to be due to hydrophobic effects and the interaction of the π face of the aromatic ring with the polarized, partially positively charged C–H bonds of the prolyl ring (*Zondlo, 2013*). Overall, hydrogen bonding, hydrophobic packing, and proline-aromatic interactions create a tight binding pocket in HypD, specifically catered to a larger cyclic, polar substrate such as Hyp.

## Site-directed mutagenesis experiments confirm that residues coordinating Hyp play critical roles in substrate stabilization and catalysis

With structural data in hand, we sought to explore the importance of active site residues through site-directed mutagenesis experiments. We first confirmed that Gly765 forms the glycyl radical species and that Cys434 is catalytically essential by constructing HypD variants containing G765A and C434S substitutions. As expected, EPR spectroscopy showed no Gly radical formation for the G765A variant (*Table 2*, *Figure 5—figure supplement 1*). Although the C434A variant can be activated by HypD-AE (*Table 2*, *Figure 5—figure supplement 1*), it had no detectable Hyp dehydration activity in our endpoint assay (*Figure 5A–C*), supporting an essential role for Cys434 in catalysis.

We next investigated residues predicted to mediate acid-base chemistry. A series of HypD variants were constructed to disrupt putative interactions with the hydroxyl group of Hyp (E436Q, H160Q, D278N), the amino group of Hyp (D278N), and the ordered water molecule (D339N) (*Figure 4E*). Detection of glycyl radical by EPR spectroscopy confirmed that substitution of these residues did not abolish activation by HypD-AE (*Figure 5—figure supplement 1*). However, these variants were all activated at substantially reduced levels, indicating these residues may participate in interactions that affect glycyl radical formation and/or stability.

**Table 2.** Glycyl radical quantification, activity, and kinetic parameters for HypD variants.

Mean and SD are displayed for glycyl radical quantification where n = 3 independent experiments for each protein. HypD activity was coupled to P5CR and absorbance at 340 nm was measured to calculate initial rates for NADH consumption. The un-normalized turnover number ($k_{cat}$) was calculated using the concentration of dimeric HypD in assays. The $k_{cat}$ was normalized by the amount of activated enzyme as determined by EPR spectroscopy. Catalytic efficiency was calculated using normalized $k_{cat}$ values.

| HypD | Radical per monomer (%) | Activity detected by quantification of proline | $K_m$ (mM) | Un-normalized $k_{cat}$ (s$^{-1}$) | Glycyl radical-normalized $k_{cat}$ (s$^{-1}$) | Catalytic efficiency using normalized $k_{cat}$ (M$^{-1}$ s$^{-1}$) |
|---|---|---|---|---|---|---|
| Wildtype (*Levin et al., 2017*) | 51 ± 1 | Yes | 1.2 ± 0.1 | 46 ± 1 | 45 ± 1 | 3.8 ± 0.3 × 10$^4$ |
| G765A | 0 | No | ND | ND | ND | |
| C434S | 34 ± 8 | No | ND | ND | ND | |
| E436Q | 12.4 ± 0.5 | No | ND | ND | ND | |
| H160Q | 4.4 ± 0.8 | No | ND | ND | ND | |
| D278N | 16 ± 4 | No | ND | ND | ND | |
| D339N | 18 ± 8 | No | ND | ND | ND | |
| S334A | 50 ± 19 | No | ND | ND | ND | |
| Y450F | 29 ± 4 | Yes | 19 ± 3 | 0.33 ± 0.01 | 0.57 ± 0.04 | 30 ± 6 |
| T645A | 19 ± 1 | Yes | 4.9 ± 0.4 | 0.75 ± 0.01 | 1.98 ± 0.04 | 400 ± 30 |
| Y450F/T645A | 3.4 ± 0.8 | No | ND | ND | ND | |
| F340A | 23 ± 5 | No | ND | ND | ND | |

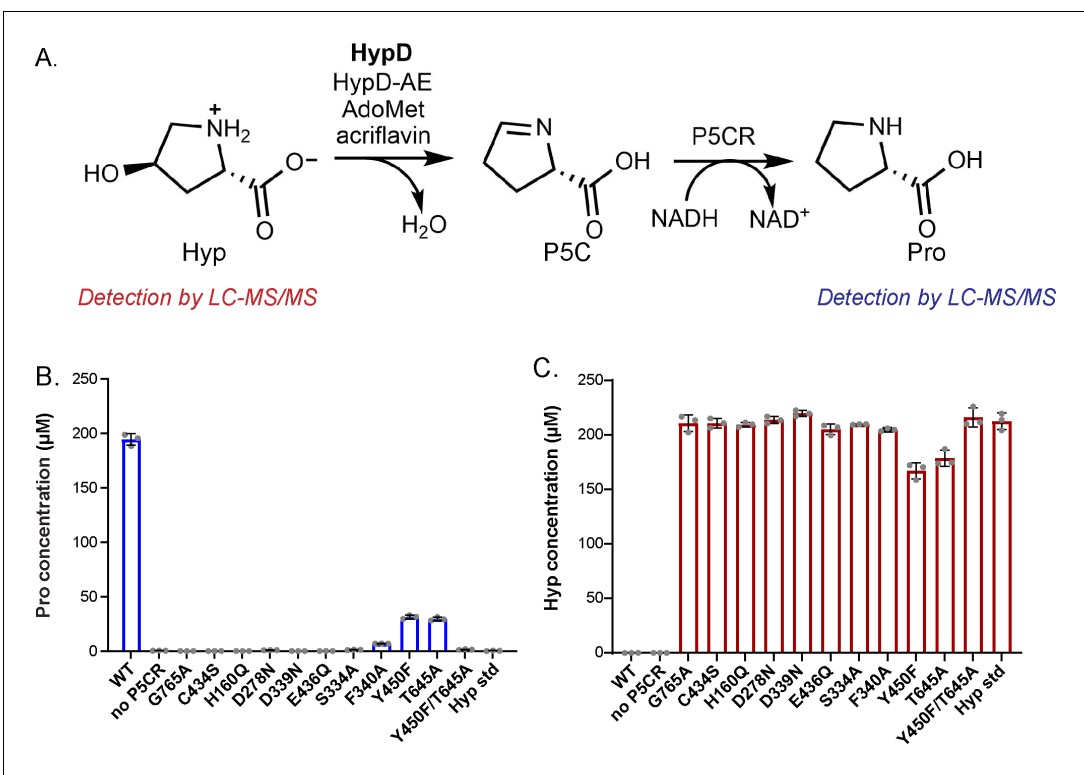

**Figure 5.** Most HypD variants do not have detectable activity. (**A**) An in vitro coupled enzyme endpoint assay was used to measure activity of HypD variants. P5C generated from HypD activity was reduced to Pro by P5CR in assay mixtures. Pro and Hyp were quantified using LC-MS/MS. (**B**) Pro concentrations in assay mixtures after incubation for 21 hr. (**C**) Hyp concentrations in assay mixtures after incubation for 21 hr. Data points represent mean ± SD with n = 3 replicates. Individual data points are displayed for each assay (n = 3 individual experiments) along with the mean and SD.

The online version of this article includes the following source data and figure supplement(s) for figure 5:

**Source data 1.** Quantification of Pro and Hyp after HypD coupled assay using LC-MS/MS.
**Source data 2.** Source data for kinetic analysis of HypD-Y450F and HypD-T645A enzyme variants.
**Figure supplement 1.** Activation of HypD variants detected by EPR spectroscopy.
**Figure supplement 2.** Kinetic analysis of HypD-Y450F and HypD-T645A.
**Figure supplement 3.** SDS-PAGE of purified proteins used in this study.

---

The D278N variant lacked Hyp dehydration activity, supporting a crucial role for Asp278 in catalysis (*Figure 5B–C*, *Table 2*). The close proximity of Asp278 to both the amino (3.1 Å) and hydroxyl (3.5 Å) moieties of Hyp suggests that this residue could both deprotonate the amino group of zwitterionic Hyp and protonate the departing hydroxyl group (*Figure 4A,E*). Either or both of these roles would explain the loss of activity in the D278N variant. Additionally, E436Q and H160Q variants lacked Hyp dehydration activity in an endpoint assay, confirming that these residues are also essential (*Figure 5B–C*, *Table 2*). Because of the close proximity of Glu436 to His160 (2.5 Å) (*Figure 4A*), we propose that His160 tunes the protonation state of Glu436, potentially allowing it to be a hydrogen bond acceptor of the hydroxyl group of Hyp (2.7 Å) (*Figure 4E*). As discussed above, this role for Glu436 would be unique in comparison to other GRE eliminases.

Strikingly, the D339N HypD variant also lacked Hyp dehydration activity (*Figure 5B–C*, *Table 2*). The lack of activity in this variant is particularly surprising because of its distance from the HypD active site (6.3 Å in the Hyp-bound structure, *Figure 4A*; 7 Å in the glycerol-bound structure, *Figure 2—figure supplement 1B*). However, the biochemical data and the absolute conservation of Asp339 among HypD sequences support a critical role for this residue.

The residues predicted to mediate acid-base chemistry (His160, Asp278, Asp339, Glu436) are part of an extensive hydrogen-bonding network surrounding the hydroxyl and amino groups of Hyp

and the conserved water molecule (*Figure 4D–E*). This network provides a plausible route for the proton transfer required to reset the protonation states of active site residues for subsequent rounds of turnover. Participation in proton transfer may contribute to the lack of activity observed when any one of these residues is altered. Notably, the presence of a similar network was also reported in CutC (*Bodea et al., 2016*). Overall, the loss of detectable activity in these variants strongly supports catalytic roles for His160, Asp278, Asp339, and Glu436.

We also altered additional HypD residues predicted to be important for substrate binding (F340A, S334A, Y450F, and T645A) (*Figure 4B*). Interestingly, although HypD-F340A was activated by HypD-AE (*Figure 5—figure supplement 1*), it displayed no catalytic activity (*Figure 5B–C*, *Table 2*). This result provides support for the proposal that a proline-aromatic interaction with Phe340 is important for Hyp binding. Ser334, Tyr450, and Thr645 are positioned within hydrogen bonding distance of the carboxylate group of Hyp (*Figure 4B*). As expected, glycyl radical formation was detected in these three variants (*Figure 5—figure supplement 1*). Strikingly, these three variants had different effects on activity. No activity was observed in the S334A variant, indicating this amino acid is essential, likely because of a role in substrate binding/positioning and/or product release. In contrast, HypD-Y450F and HypD-T645A exhibited drastically reduced activity (*Figure 5B–C*, *Table 2*), suggesting that they may participate in important interactions but are not essential.

Kinetic assays were performed for HypD-Y450F and HypD-T645A to examine effects of these mutations on catalysis. The catalytic efficiencies for each HypD variant were 2–3 orders of magnitude lower than that of wildtype HypD due to increased $K_m$ and decreased $k_{cat}$ values (*Table 2*, *Figure 5—figure supplement 2A–B*). Given that Tyr450 and Thr645 participate in hydrogen bonding interactions with the carboxylate group, increased $K_m$ values support a role for these residues in substrate binding. Decreases in $k_{cat}$ could be explained by changes in Hyp orientation that might reduce the catalytic rate due to suboptimal distances between Hyp and the catalytic residues. Finally, conformational changes caused by these mutations may lead to protein destabilization and thus could also contribute to a reduction in $k_{cat}$. Since both Tyr450 and Thr645 interact with the same carboxylate oxygen atom of substrate, we generated a double-mutant variant, HypD-Y450F/T645A, to test if these two residues have redundant functions. HypD-Y450F/T645A was activated by HypD-AE as detected by EPR spectroscopy (*Figure 5—figure supplement 1*) but showed no detectable activity in the endpoint assay (*Figure 5B–C*, *Table 2*).

## Experiments with deuterated Hyp suggest that a hydrogen atom is transferred from C5 to C4 of Hyp during catalysis

We next investigated whether the deuterated substrate, [2,5,5-D3]-*trans*-4-hydroxy-L-proline (2,5,5-D3-Hyp), could be used as a mechanistic probe in a coupled assay with P5CR (*Figure 6A*) to study the radical-based reaction of HypD. Concerned about the potential issue of deuterium wash-out from reaction intermediates in the coupled assay, we first conducted a control experiment. Nonenzymatic, hydrolytic ring-opening and tautomerization of the HypD product P5C (*Figure 6B*) and solvent accessibility of Cys434 could both lead to substantial deuterium wash-out for a reaction performed in $H_2O$. To address these potential pitfalls, we performed the HypD coupled assay in $D_2O$ using unlabeled Hyp, reasoning that if ring-opening and tautomerization presented a major problem, and/or if Cys434 was solvent exposed during catalysis, we would see deuterium incorporation in the final product of the coupled assay, proline. Remarkably, we observed little deuterium incorporation in product (~10% based on the assay run in triplicate and compared to a control of Pro dissolved in $D_2O$) (*Figure 6C*, see *Figure 6—source data 1* for relevant MS/MS transitions). These data indicate that solvent accessibility of Cys434 is not a major issue and that if P5C ring-opening/tautomerization is occurring, it is not happening quickly enough to have a substantial impact on P5C hydrogen/deuterium exchange.

We next performed the HypD coupled assay with 2,5,5-D3-Hyp as the substrate, and formation of [2,4,5-D3]-L-proline was observed, characterized by high-resolution MS, LC-MS/MS and [1]H, [13]C, [1]H-[1]H COrrelated SpectroscopY (COSY), and [1]H-[13]C heteronuclear single quantum coherence ([1]H-[13]C HSQC) NMR experiments (*Figure 6D–E*, *Figure 6—figure supplements 1–3*). Thus, we observe a shift of a deuterium atom from C5 to C4, which is consistent with C5 being the site of hydrogen-atom abstraction and C4 being the site of hydrogen-atom re-abstraction. A potential mechanism consistent with this experimental result is described below.

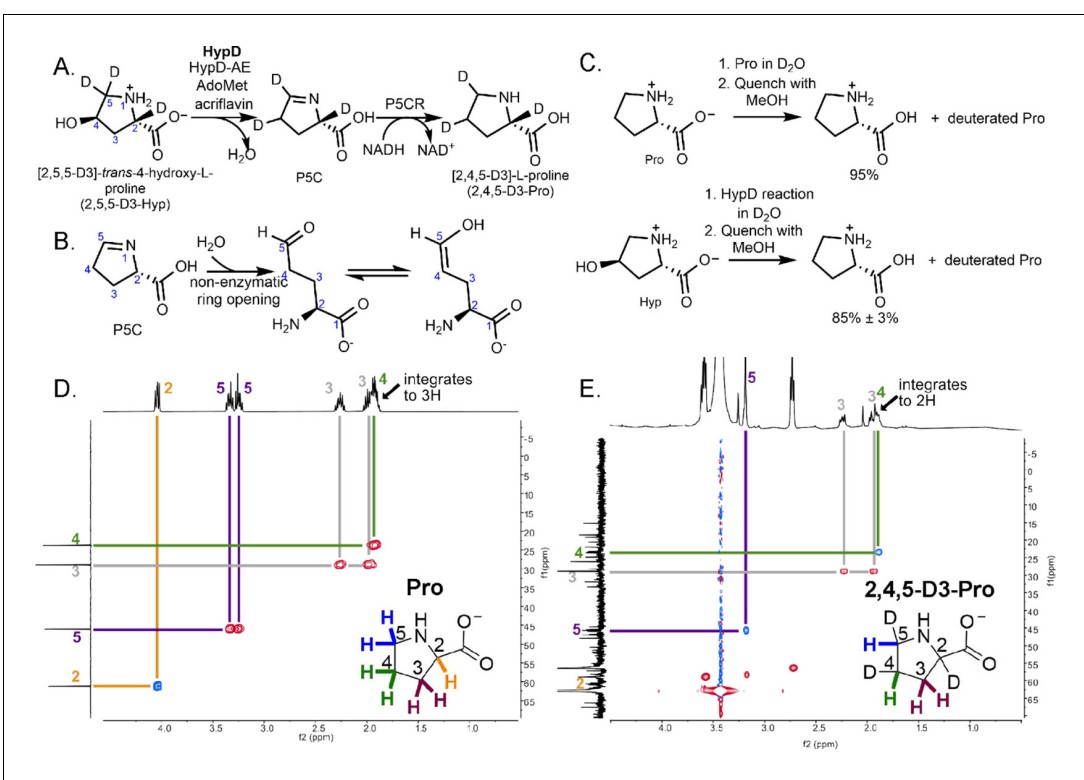

**Figure 6.** Coupled HypD and P5CR assay with trideuterated substrate 2,5,5-D3-Hyp results in the formation of product 2,4,5-D3-Pro. (**A**) Overall reaction scheme for HypD P5CR coupled assay with 2,5,5-D3-Hyp. (**B**) P5C nonenzymatically hydrolyzes to an aldehyde product that equilibrates between keto and enol tautomers. (**C**) HypD assays in $D_2O$ were performed in triplicate, and products were detected by LC-MS/MS. The percentage of undeuterated Pro is indicated. As a control, LC-MS/MS was also performed on Pro in $D_2O$ to determine the extent of deuteration resulting from solvent exchange. (**D**) $^1H$-$^{13}C$ HSQC NMR of a commercial standard of Pro. Cross peaks between $^{13}C$ (*f1*, y-axis) and $^1H$ (*f2*, x-axis) are indicated by color-coded lines matching the inset Pro molecule. (**E**) $^1H$-$^{13}C$ HSQC NMR of the product of HypD assay using 2,5,5-D3-Hyp as substrate. HSQC spectra were multiplicity edited. Red cross peaks correspond to $CH_2$ signals, and blue cross peaks correspond to CH signals. Notably, compared to panel D, the product shows only one H at position C4, suggesting deuterium incorporation at this position. Furthermore, only one H is bonded to C5 showing that one deuterium atom was lost at this position during catalysis.

The online version of this article includes the following source data and figure supplement(s) for figure 6:

**Source data 1.** LC-MS/MS data for HypD $D_2O$ assay and HypD assay using 2,5,5-D3-Hyp as substrate.
**Figure supplement 1.** $^1H$ NMR of Pro and HypD coupled assay product 2,4,5-D3-Pro.
**Figure supplement 2.** $^{13}C$ NMR of Pro and HypD coupled assay product 2,4,5-D3-Pro.
**Figure supplement 3.** COrrelated SpectroscopY (COSY) NMR of HypD coupled assay product 2,4,5-Pro.

## Discussion

Experimental and computational investigations have provided strong support for direct elimination as the mechanism used by GRE eliminases (*Feliks and Ullmann, 2012*; *Kovačević et al., 2018*; *Levin and Balskus, 2018*). On the basis of these previous studies and the structural insights gained in this work, we propose a possible mechanism for HypD, similar to those proposed for other GRE eliminases, that involves a direct elimination of the hydroxyl group on Hyp to generate P5C (*Figure 7*).

After activation of HypD by HypD-AE, we predict that Hyp binds as a zwitterion in the active site based on its protonation state at neutral pH (*Figure 7*, step I). The positive charge of the Hyp amino group is likely stabilized by a cation-π interaction with Phe340 and hydrogen bonding interactions with Asp292 and solvent. The Hyp carboxylate is anchored by hydrogen bonding with Ser334,

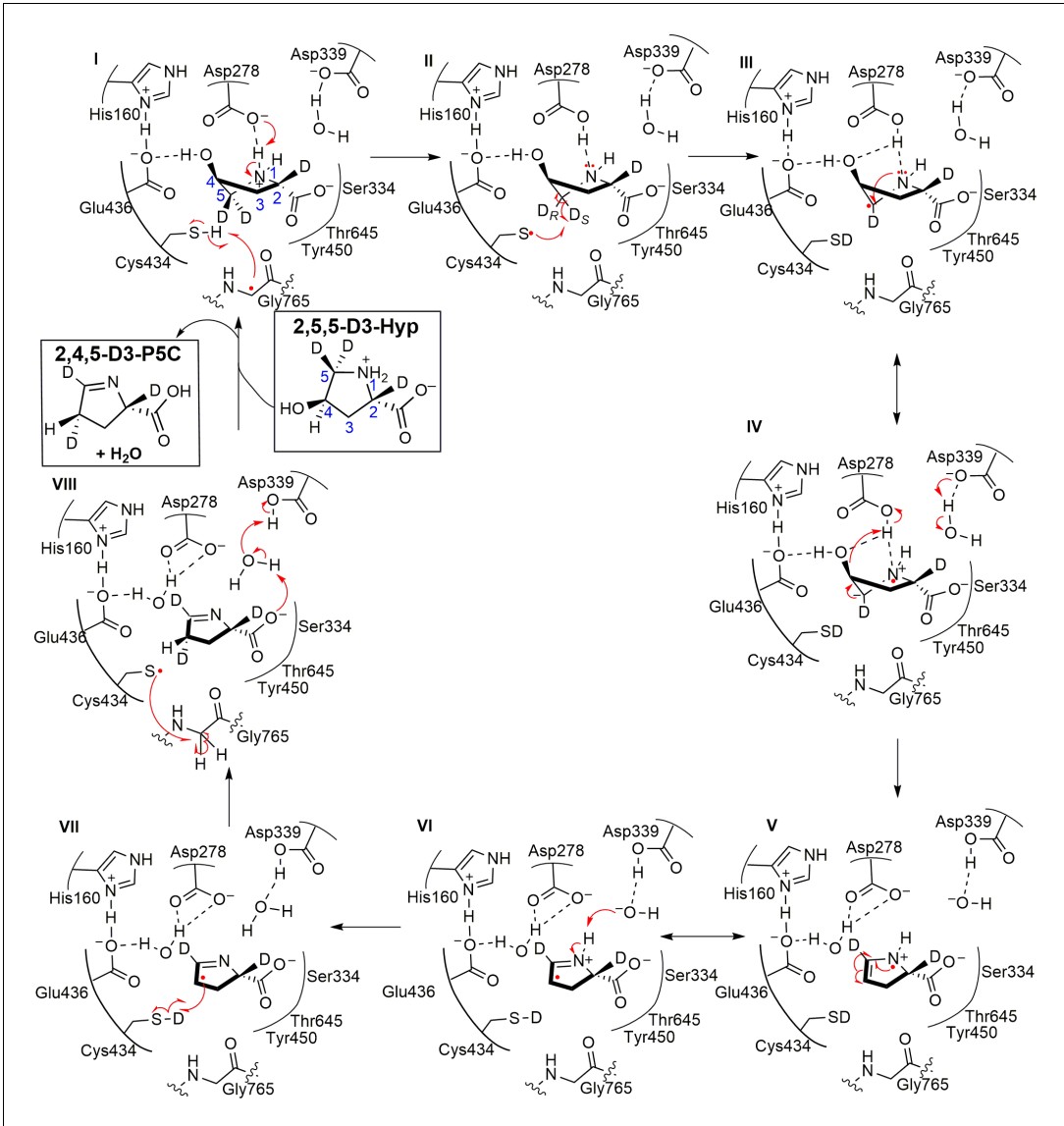

**Figure 7.** Proposed mechanism for HypD dehydration of deuterated substrate 2,5,5-D3-Hyp.

Tyr450, Thr645, and Asp339 (through a water molecule), with its negative charge counteracted by nearby positively charged residues Lys326 and Arg156 (*Figure 4D*).

Once the radical has been transferred to the nearby Cys434 (*Figure 7*, step I), the transient thiyl radical can then abstract a hydrogen atom from the substrate. Based on the biochemical assay with 2,5,5-D3-Hyp and our structural data, we propose that hydrogen atom abstraction occurs at the C5 carbon (*Figure 7*, step II). In particular, the structure suggests that Cys434 abstracts the *pro-S* hydrogen atom of C5, which is the closest to Cys434 (2.6 Å) (*Figure 8A*). Hydrogen atom abstraction of the *pro-S* instead of the *pro-R* hydrogen atom is consistent with the stereochemistry determined for the reaction catalyzed by PFL (*Frey et al., 1994*) and that is proposed for most other GRE eliminases based on structural data (*O'Brien et al., 2004*; *LaMattina et al., 2016*; *Xing et al., 2019*; *Bodea et al., 2016*). Since the initial hydrogen atom abstraction steps in GRE-catalyzed transformations have high activation energy barriers (*Feliks and Ullmann, 2012*; *Liu et al., 2010*; *Yang et al., 2019*), we propose that deprotonation of the Hyp amino group occurs first, as the lone pair of a free amino group could stabilize the radical forming on C5 through conjugative electron delocalization

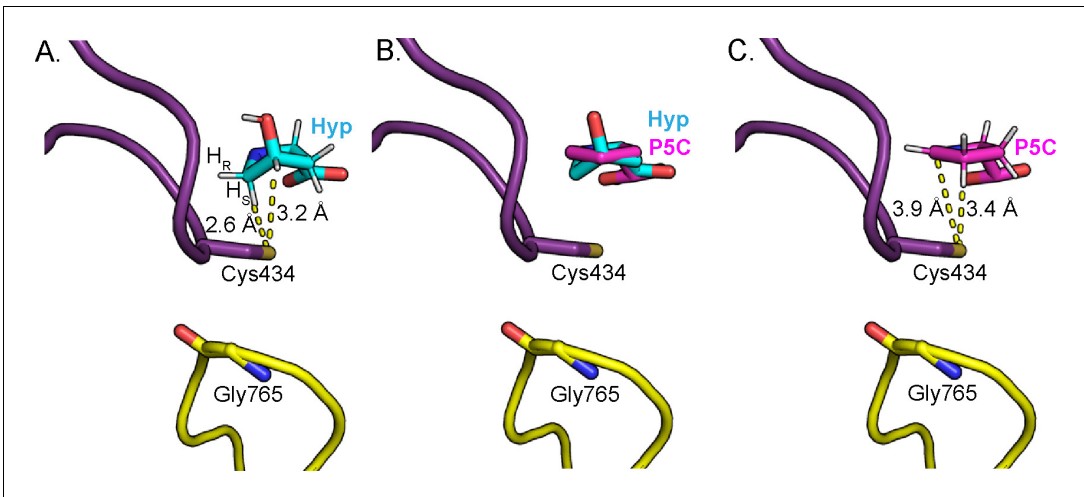

**Figure 8.** Change in Hyp puckering throughout the mechanism is proposed to play a critical role in radical transfer. (**A**) $C^{\gamma}$-*exo* puckering of Hyp positions the *pro*-S hydrogen atom of Hyp C5 in closest proximity to Cys434 for hydrogen atom abstraction. (**B**) The product P5C is modeled into Hyp-bound HypD structure by manual docking, such that carboxylate tails of both P5C and Hyp are anchored in the same location. (**C**) The pyrroline ring of P5C exhibits a planar structure, positioning C4 closer to Cys434 than C5 for hydrogen abstraction from the thiyl group. .

(*Figure 7*, step I to II) (*Viehe et al., 1985*). The structure suggests that Asp278 is positioned appropriately to act as the catalytic base that deprotonates the amino group (*Figure 7*, step I). Indeed, activity assays confirm that Asp278 is essential. Here we show deprotonation of the amino group of Hyp preceding hydrogen atom abstraction from C5 (*Figure 7*, step I to II).

Hydrogen atom abstraction from C5 of Hyp results in an α-aminoalkyl radical intermediate (*Figure 7*, step II to III). α-Aminoalkyl radicals are most often observed in amino acid-based intermediates generated during GRE activation, amino acid epimerization, and thioether bond formation in biosynthesis of ribosomally synthesized and post-translationally modified peptides (RiPPs) (*Benjdia et al., 2017*). Although $C^{\alpha}$-centered radicals of amino acids are stabilized by delocalization of the unpaired electron onto adjacent amino and carboxyl groups (the captodative effect) (*Viehe et al., 1985*), HypD presents a unique case where an α-aminoalkyl radical intermediate forms in the absence of an adjacent carboxyl group. Although rarely reported in enzymatic chemistry, α-aminoalkyl radicals are often generated as reactive intermediates to drive substitution reactions in organic chemistry (*Renaud and Giraud, 1996*; *Nakajima et al., 2016*; *Mondal et al., 2014*). Given their wide utility to synthetic chemists, it is tempting to speculate that these radicals are more common among enzymatic reactions than has been previously appreciated.

A direct elimination of the hydroxyl group on C4 is expected to proceed from the α-aminoalkyl radical species, which is in resonance with the corresponding aminyl radical (*Figure 7*, resonance structures III and IV). We propose that rapid elimination of the hydroxyl leaving group from this intermediate occurs and is facilitated by protonation of the departing hydroxyl group by a catalytic acid (*Figure 7*, step IV to V). It is also possible that this elimination step occurs in a concerted manner with N-deprotonation (*Figure 7*, step VI to VII). Based on the structure and mutagenesis, we propose that the catalytic acid is Asp278, the same residue that likely deprotonates the Hyp amino group, which would reset Asp278's protonation state (*Figure 7*, step II to IV). The structure also suggests that His160 and Glu436 are likely to be important in positioning the hydroxyl group of Hyp for protonation and elimination (*Figure 7*, step IV). Whereas the Glu residue of the CXE motif is generally believed to act as the catalytic base in other GRE eliminase mechanisms to generate ketyl radical intermediates (*Figure 3—figure supplement 3*; *O'Brien et al., 2004*; *LaMattina et al., 2016*; *Xing et al., 2019*; *Bodea et al., 2016*), here the role of Glu436 appears to be unique, but not less important. Mutagenesis experiments showed that Glu436, along with Asp278 and His160, are essential (*Figure 5*).

The elimination of the hydroxyl group of Hyp generates a double bond, resulting in the generation of an enamine radical cation intermediate (*Figure 7*, step V). Because the HypD coupled assay with 2,5,5-D3-Hyp resulted in formation of 2,4,5-D3-Pro (*Figure 6D–E*), suggesting that the initial deuterium atom abstracted from C5 by Cys434 is transferred back to C4, we propose that the final radical transfer from P5C to Cys434 occurs at C4 (*Figure 7*, step VII). This proposal also makes sense from a structural perspective. Formation of this double bond settles the Hyp's $C^\gamma$-*exo* pucker into a nearly planar pyrroline species. Manual docking of the product P5C into the active site (*Figure 8B*) allows us to approximate how this change in ring puckering should affect the relative proximities of substrate/product atoms to the thiyl radical-forming Cys434 (*Figure 8*). Specifically, modeling suggests that C4 of P5C is closer to Cys434 than is C5 (*Figure 8B*).

Formation of such a C4 radical species on product will be facilitated by resonance delocalization of the unpaired electron on the nitrogen atom across the adjacent alkene (*Figure 7*, resonance structures V and VI). Hydrogen atom re-abstraction from Cys434 could occur at this point or may follow the deprotonation of an enamine radical cation intermediate ($pK_a$ ~5.5–7 *Jonsson et al., 1996*; *Wang et al., 2017*), which would be expected to generate a more nucleophilic, α-iminyl radical species (*Figure 7*, step VI to VII) (*Roberts, 1999*). Asp339, which we show here is catalytically essential, appears well positioned to serve as the catalytic base via a water-mediated interaction (*Figure 7* step VI to VII). Asp339 could also serve as a catalytic acid in the subsequent protonation of the P5C carboxylate (*Figure 7*, step VII to VIII). Protonation of the P5C carboxylate should aid in its release as the very close interaction with the P5C carboxyl group and Ser334 (2.5 Å) would be unfavorable if the carboxyl moiety were protonated. The final mechanistic step is reformation of the glycyl radical species on HypD (*Figure 7*, step VIII) and product release.

Importantly, these structural and biochemical data have allowed us to identify a key set of HypD residues important for catalysis and substrate binding, leading to a deeper understanding of how this prominent human gut bacterial enzyme performs difficult radical chemistry. Not only will these data inform drug design efforts directed toward *C. difficile* HypD, they should also allow us to more accurately identify other pathogens that contain HypD enzymes. The Hyp dehydration is an exciting transformation that is unique among GREs and more broadly among enzymes. Herein we propose the first mechanism for chemical reversal of one of the most common post-translational modifications, proline hydroxylation. Through HypD, gut microbes have found a way to opportunistically profit from the abundant yet catabolically underutilized metabolite Hyp.

## Materials and methods

All chemicals, solvents, and reagents were purchased from Sigma-Aldrich unless otherwise noted. Luria-Bertani Lennox (LB) medium was purchased from EMD Millipore or Alfa Aesar. DNA sequencing results and multiple sequence alignments were analyzed with Geneious 9.0.4 (*Kearse et al., 2012*). Primers were purchased from Integrated DNA Technologies (Coralville, IA) or Sigma-Aldrich. PCR was performed with a C1000 Gradient Cycler (Bio-Rad). All plasmid constructs were verified by DNA sequencing through Eton Biosciences. All restriction enzymes, ligases, polymerases, and PCR mixes were obtained from New England Biolabs. Protein solutions were routinely denatured at 90°C for 10 min in equal volume Laemmli sample buffer (BioRad) prior to visualization by SDS-PAGE (4–15% Tris-HCl gel, Bio-Rad). 2-Mercaptoethanol was added at a final concentration of 355 mM to Laemmli sample buffer. Fractions from protein purifications were routinely visualized by SDS-PAGE following staining (Biosafe Coomassie, Bio-Rad). Isopropyl β-D-1-thiogalactopyranoside (IPTG) was obtained from Teknova. Ni-NTA and TALON resin were obtained from Qiagen and Clontech, respectively. All absorbance measurements in 96-well plates were carried out using a PowerWave HT Microplate Spectrophotometer (Biotek) inside of an anaerobic chamber (MBraun). All absorbance data shown for kinetic assays were obtained with pathlength corrected to 1 cm. All enzyme assays were carried out in an MBraun anaerobic chamber.

Samples were made anaerobic as follows. Solids were brought into the anaerobic chamber (MBraun) in perforated 1.7 mL microcentrifuge tubes. Protein solutions with volumes greater than 1 mL were made anaerobic on a Schlenk line by cycling between vacuum and argon. Buffers and other solutions were rendered anaerobic in 1 to 20 mL volumes by bubbling argon or nitrogen through the liquid for about 30 min.

## Vector constructs for the overexpression of HypD, HypD-AE, and P5CR

HypD (UniProt ID: A0A031WDE4), HypD-AE (UniProt ID: A0A069AMK2), and P5CR (UniParc ID: UPI000235AE56) genes were amplified from *C. difficile* 70-100-2010 genomic DNA to construct expression vectors pET28a-CdHypD, pET28a-CdP5CR, and pSV272-PfMBP-CdHypD-AE as previously reported (*Levin et al., 2017*).

## Overexpression and purification of proteins used for biochemical experiments

All proteins used in biochemistry studies were overexpressed and purified as previously reported (*Levin et al., 2017*). Briefly, all proteins were overexpressed in *E. coli* BL21-CodonPlus(DE3)-RIL *ΔproC::aac(3)IV* (apramycin resistant or AmR) and induced with 0.1 mM IPTG (HypD and HypD-AE) or 0.5 mM IPTG (P5CR). Proteins were purified and then dialyzed into buffer containing 25 mM Tris buffer pH 7.5, 50 or 100 mM KCl, 5 mM DTT to remove imidazole used for elution. HypD and HypD-AE were purified using TALON metal affinity resin while P5CR was purified using Ni-NTA resin. HypD-AE was rendered anaerobic using a Schlenk line for reconstitution of [4Fe-4S] clusters. Briefly, HypD-AE was incubated with 10 mM DTT, 12 equiv $Na_2S \cdot 9\ H_2O$ to protein, and 12 equiv $Fe(NH_4)_2(SO_4)_2 \cdot 6\ H_2O$ to protein for 12 hr. Reconstituted protein was buffer exchanged into 25 mM Tris buffer pH 7.5, 100 mM KCl, 5 mM DTT for storage.

All protein solutions were made anaerobic prior to freezing and storage at –80°C. Protein concentrations were calculated using Abs280 measurements from Nanodrop as previously reported for wild-type enzymes. A molar extinction coefficient of 80,680 $M^{-1}\ cm^{-1}$ was used for Y450F and Y450F/T645A variants, and 82,170 $M^{-1} cm^{-1}$ was used for remaining HypD variants. All proteins were visualized by denaturing polyacrylamide gel electrophoresis to confirm high purity (*Figure 5—figure supplement 3*).

## Overexpression and purification of HypD for crystallography studies

A frozen glycerol stock of *E. coli* BL21-CodonPlus(DE3)-RIL transformed with pET28a-CdHypD vector was streaked onto an LB-agar plate containing 50 µg $mL^{-1}$ kanamycin. A 50 mL starter culture in LB media and a monoclonal culture from this agar plate were incubated shaking at 37°C overnight. Aliquots of the overnight culture were used to inoculate four 1 L cultures of LB media containing 50 µg $mL^{-1}$ kanamycin in Corning Erlenmeyer Baffled cell culture flasks. The cultures were incubated shaking at 37°C until $OD_{600}$ ~0.70, at which point protein overexpression was induced with 1 mM IPTG. Incubation continued for an additional 16 hr at 25°C. Cells were pelleted and lysed with a buffer containing 20 mM HEPES pH 8.0, 100 mM NaCl, 0.5 mM TCEP, and 4.25 mg of lysozyme (Sigma-Aldrich). Soluble, overexpressed HypD was purified from clarified lysate using Ni-NTA resin, with HypD eluting at 300 mM imidazole. HypD was buffer exchanged into 20 mM HEPES buffer pH 8.0, 100 mM NaCl, and 0.5 mM TCEP to remove imidazole, yielding approximately 10 mg of HypD, calculated using a coefficient of 78,300 $M^{-1} cm^{-1}$, determined using the ProtParam tool (*Wilkins et al., 1999*). All protein solutions were frozen and stored at –80°C. Protein was visualized by sodium dodecyl sulfate polyacrylamide gel electrophoresis (SDS-PAGE) to confirm high purity (*Figure 5—figure supplement 3*).

## Crystallization of HypD

All crystallization experiments were performed aerobically with unactivated wild-type HypD protein with the intact N-terminal hexahistidine tag. Initial screening was performed with the aid of an Art Robbins Phenix micro-pipetting robot and Formulatrix Rock Imager, and initial conditions were found using the Qiagen Protein Complex screen, with optimization yielding a well solution containing 14% (w/v) polyethylene glycol (PEG) 3350, 100 mM potassium chloride, and 100 mM HEPES pH 7.5. Diffraction-quality crystals were optimized in hanging drop vapor diffusion trays at 21°C. Protein at 100 µM (9 mg $mL^{-1}$) in buffer containing 20 mM HEPES buffer pH 8.0, 100 mM NaCl, and 0.5 mM TCEP was mixed with well solution in a 1:1 ratio. Plate-like crystals formed and grew to maximum size after 1–2 days of equilibration. Crystals were cryoprotected by soaking for 1 min in solution containing 15% (v/v) glycerol, 14% (w/v) PEG 3350, 100 mM potassium chloride, and 100 mM HEPES pH 7.5. The following modification was made for the crystallization of substrate-bound HypD: 4 mM Hyp was added to the crystallization buffer prior to mixing with protein at a 1:1 ratio. The 15% (v/v)

glycerol was excluded in the cryoprotectant solution in which 25% (v/v) dimethyl sulfoxide (DMSO) and 100 mM Hyp were added.

## Structure determination of HypD

Crystals were indexed in space group $P2_1$, and diffraction images were collected at the Advanced Photon Source beamline 24ID-E at a wavelength of 0.9795 Å on a Pilatus 6M detector (Dectris) (*Table 1*). Data were indexed, integrated, and scaled in XDS (*Kabsch, 2010a*; *Kabsch, 2010b*). The first structure obtained for HypD contained a glycerol bound in the active site and was solved by molecular replacement in the Phenix implementation of Phaser (*McCoy et al., 2007*). CutC was used as a search model (PDB: 5FAU) (*Bodea et al., 2016*) after trimming of side chains in Phenix Ensembler (*Adams et al., 2010*). A solution with eight molecules per asymmetric unit was found with an initial $R_{free}$ of 0.52 at 2.05 Å resolution. Several rounds of initial refinement in phenix.refine (*Adams et al., 2010*) with tight NCS restraints and optimization of group B factors were sufficient to reduce $R_{free}$ values below 0.4 (*Table 1*). Subsequently modeling in side chains and eventually modeling in water molecules, further reduced $R_{free}$ to ~0.3. NCS restraints were removed after initial refinement. Positional and individual B-factor refinement continued at the full resolution until the model was complete.

The Hyp-bound HypD structure was later solved using this glycerol-bound HypD structure as a molecular replacement model. Initial molecular replacement of the Hyp-bound HypD structure resulted in an initial $R_{free}$ of 0.31 at 2.52 Å resolution (*Table 1*). Simulated annealing was performed after molecular replacement to decrease biases from the glycerol-bound structure. Positional and individual B-factor refinement continued at the full resolution until the model was complete (*Table 1*). Hyp was fit into difference electron densities and verified with simulated annealing composite omit maps in all active sites of molecules in the asymmetric unit. Substrate constraints were calculated by density function theory (DFT) as described below. Parameter files for Hyp were generated in Phenix eLBOW (*Moriarty et al., 2009*). Water molecules were placed automatically after ligands were refined and verified manually. No density is observed for the hexahistidine affinity tag in either the glycerol or Hyp bound structures. However, all other residues can be visualized. Structural figures were made in PyMOL v2.0.7 (The PyMOL Molecular Graphics System, Version 2.0 Schrodinger, LLC). Crystallography software packages were compiled by SBGrid (*Morin et al., 2013*).

## Density function theory calculations for Hyp conformation

No small molecule structure of *trans*-4-hydroxy-L-proline has been reported. Thus, we calculated the energies associated with the two different puckering possibilities of Hyp, C$^{\gamma}$-*exo* and C$^{\gamma}$-*endo*, for modeling of Hyp in the refinement of the Hyp-bound crystal structure. Density Function Theory (DFT) calculations were performed for the zwitterionic Hyp using B3LYP/6–31G* theory using Gaussian 16 (*Frisch, 2016*) in the gas phase (*Figure 2—figure supplement 2*, *Figure 2—source datas 1–2*). The zwitterionic structure was enforced by freezing the N–H bond length during optimization. Without this constraint, the proton transferred to the carboxylate during optimization. These calculations revealed that the C$^{\gamma}$-*exo* pucker is more energetically favorable than the C$^{\gamma}$-*endo* pucker by 2.7 kcal mol$^{-1}$. Energies were also calculated for the anionic Hyp containing a neutral amine. Similarly, the C$^{\gamma}$-*exo* pucker was found to be more energetically favorable by 2.9 kcal mol$^{-1}$. Furthermore, deprotonation of the amino group did not significantly affect the conformation of each pucker. Based on these results, zwitterionic Hyp in the C$^{\gamma}$-*exo* pucker state was used to generate a parameter file for the crystallographic refinement. Cartesian coordinates for the *exo* pucker are listed in *Figure 2—source data 1* and for the *endo* pucker in *Figure 2—source data 2*.

## Site-directed mutagenesis and construction of overexpression vectors for HypD variants

Single residue mutations were introduced in pET28a-CdHypD through site-directed mutagenesis using the corresponding primers listed in *Table 3*. The following residue changes were made: H160Q, D278N, S334A, D339N, F340A, C434S, E436Q, Y450F, T645A, G765A. A double-mutation variant for CdHypD was constructed by introducing Y450F mutation into the vector pET28a-CdHypD-T645A. PCR was carried out using Phusion-HF or Q5 polymerase according to the

**Table 3.** Primers used in site-directed mutagenesis of HypD.
Nucleotides mutated are indicated in small letters.

| Primer | Sequence (5′ to 3′) | Annealing temperature used, ˚C |
|---|---|---|
| pET28a-CdHypD-G765A-fwd | GACTTAATAGTTAGAGTTGCAGcATATAGTGACCATTTC | 66 |
| pET28a-CdHypD-G765A-rev | CTACTTAAATTATTGAAATGGTCACTATATgCTGCAACTCTAAC | 66 |
| pET28a-CdHypD-C434S-fwd | AACCAGTGGTTcTGTTGAAACTGGATG | 58 |
| pET28a-CdHypD-C434S-rev | CAGTTTCAACAgAACCACTGGTTCCACC | 58 |
| pET28a-CdHypD-E436Q-fwd | CAGTGGTTGTGTTcAAACTGGATGTTTTGG | 60 |
| pET28a-CdHypD-E436Q-rev | ACATCCAGTTTgAACACAACCACTGGTTC | 60 |
| pET28a-CdHypD-H160Q-fwd | AGCCCCAGGACAgACAGTTTGTGGAGATAC | 60 |
| pET28a-CdHypD-H160Q-rev | ACAAACTGTcTGTCCTGGGGCTCTTTGTTC | 60 |
| pET28a-CdHypD-D278N-fwd | GAACTTAATATATGGaATGCTTTTACTCCAGGAAGACTTGACC | 66 |
| pET28a-CdHypD- D278N-rev | CCTGGAGTAAAAGCATtCCATATATTAAGTTCAGTAGTAACCCC | 66 |
| pET28a-CdHypD- F340A-fwd | GAAAGTAGCACATATACAGATgcTGCAAATATAAAC | 54 |
| pET28a-CdHypD- F340A-rev | GATTTATTCCACCAGTGTTTATATTTGCAgcATCTGTATATG | 54 |
| pET28a-CdHypD- Y450F-fwd | GTTTTGGTAAAGAAGCATATGTTCTAACTGGATtTATGAACATTCC | 66 |
| pET28a-CdHypD- Y450F-rev | GTATTTTTGGAATGTTCATAaATCCAGTTAGAACATATGCTTCTTTACC | 66 |
| pET28a-CdHypD- S334A-fwd | GTTGGTATAACATTAAAAGAAgcTAGCACATATACAGATTTTGC | 60 |
| pET28a-CdHypD- S334A-rev | CTGTATATGTGCTAgcTTCTTTTAATGTTATACCAACTTTTGG | 60 |
| pET28a-CdHypD- T645A-fwd | ATGTTACCAgCAACTTGTCATATATACTTTGGAGAAATTATGGG | 66 |
| pET28a-CdHypD- T645A-rev | TATGACAAGTTGcTGGTAACATATCTACTCTGTATTCTCCACC | 66 |
| pET28a-CdHypD- D339N-fwd | CATTAAAAGAAAGTAGCACATATACAaATTTTGCAAATATAAACACTGG | 66 |
| pET28a-CdHypD- D339N-rev | GGATTTATTCCACCAGTGTTTATATTTGCAAAATtTGTATATGTGCTAC | 66 |

manufacturer's protocol in a total reaction volume of 25 µL. An extension time of 200 s at 72˚C was used for the PCR protocol, and the annealing temperatures used for each primer pair is listed in *Table 3*. Template plasmid was removed by digesting with DpnI (NEB) in all PCR mixtures at 37˚C for 1 hr. 2 µL of each PCR digestion was used to transform chemically competent *E. coli* TOP10 cells. Sequenced vectors were then transformed into chemically competent *E. coli* BL21-CodonPlus(DE3)-RIL Δ*proC::aac(3)IV* (AmR) cells for protein overexpression.

## Glycyl radical detection and quantification by EPR spectroscopy

HypD wild-type and variants were activated using HypD-AE under previously published assay conditions (*Levin et al., 2017*). Briefly, 60 µM HypD-AE was first incubated with 0.1 mM acriflavine for 20 min in buffer (20 mM Tris-HCl pH 7.5, 100 mM KCl, 50 mM bicine). 15 µM HypD and 1.5 mM Ado-Met were then added to the solution and incubated for 2 hr. Samples at both steps were placed at about 10 inches from the MBraun chamber light. The entire volume of 220 µL for each sample was used for glycyl radical quantification. All activation assays were performed in triplicate and quantified for glycyl radical content using EPR spectroscopy.

Perpendicular mode X-band EPR spectra were recorded on a Bruker ElexSysE500 EPR instrument fitted with a quartz dewar (Wilmad Lab-Glass) for measurements at 77 K. All samples were loaded into EPR tubes 4 mm in outer diameter and 8 inch in length (Wilmad Lab-Glass, 734-LPV-7), sealed, and frozen in liquid nitrogen. Data acquisition was performed with Xepr software (Bruker). The magnetic field was calibrated with an external standard of α,γ-bisdiphenylene-β-phenylallyl (BDPA), $g = 2.0026$ (Bruker). X-band EPR spectroscopy modeling and spin concentration calculations were carried out as previously described using EasySpin (Version 5.0.22) on MATLAB (MathWorks) (*Stoll and Schweiger, 2006*). An external standard of $K_2(SO_3)_2NO$ was prepared under anaerobic

conditions in 0.5 M $KHCO_3$ for each experiment. Standard concentration was calculated using absorbance at 248 nm ($\varepsilon$ = 1,690 $M^{-1}$ $cm^{-1}$) measured using a NanoDrop 2000 UV-Vis Spectrometer (*Li and Ritter, 1953*). EPR spectra represent the average of 1 to 15 scans. Two to three spectra were obtained for each assay mixture as technical replicates and were recorded under the following conditions: temperature, 77 K; center field, 3350 Gauss; sweep width, 200 Gauss; microwave power, 20 μW; microwave frequency, 9.45 MHz; modulation amplitude, 0.4 mT; modulation frequency, 100 kHz; time constant, 20.48 ms; conversion time, 20.48 ms; scan time, 20.97 s; receiver gain, 60 dB (for enzymatic assays) or 30 dB (for standards). Normalization for the difference in receiver gain was performed by the spectrometer. All EPR assays with wild-type and variants were performed in triplicate.

## End-point activity assays with HypD variants

All assays were prepared as previously described *Levin et al. (2017)*. Briefly, assays contained 20 mM Tris-HCl pH 7.5, 100 mM KCl, 0.8 mM NADH, 3 μM P5CR, 0.2 mM Hyp, and 0.3 μM HypD. HypD was first activated under conditions described for EPR spectroscopic assays. All assays were carried out in triplicate and were initiated by adding Hyp into reaction mixtures, which were then incubated for 21 hr at 22°C. Upon removal from the anaerobic chamber, reactions were quenched with a 2 × volume of methanol and protein precipitates were removed by centrifugation (15,200 g, 10 min). Supernatants were further diluted 30-fold with water for Pro detection and 7.5-fold for Hyp detection by LC-MS/MS using previously published methods (*Levin et al., 2017*). Briefly, LC-MS/MS analysis of Pro and Hyp were performed on an Agilent 6410 Triple Quadrupole LC-MS instrument (Agilent Technologies) using a Luna SCX column (5 μm, 100 Å, 50 × 2.0 mm, Phenomenex). Precursor and product ions of *m/z* 116.1 and *m/z* 70.1 were monitored for proline detection whereas *m/z* 132.1 and *m/z* 86.1 were monitored for hydroxyproline. Amino acid standards were dissolved in water and diluted to a range of concentrations to generate standard curves used to quantify Pro and Hyp in samples. Source data can be found in *Figure 5—source data 1*.

## Kinetic analysis of HypD variants using a coupled spectrophotometric assay

Activated HypD variants were used for coupled enzyme kinetic assays as previously described with a few modifications (*Levin et al., 2017*). All kinetic assays contained 20 mM Tris-HCl pH 7.5, 50 mM bicine pH 7.5, 100 mM KCl, and 400 μM NADH. 3 μM of HypD-Y450F (total monomer) and 0.75 μM of HypD-T765A (total monomer) were used along with 2 × concentration of P5CR for each assay. Assays were initiated by addition of Hyp to a final concentration of 0, 1, 2, 5, 10, 15, 30, and 60 mM. Data points represent mean ± SD for n = 3 individual experiments. The data was fit simultaneously to the Michaelis-Menten equation using nonlinear regression in Graphpad Prism 7.00. The $k_{obs}$ parameter was calculated based on 29 ± 4% (mean ± SD) activation of HypD-F340A and 19 ± 1% activation of HypD-T645A as determined by EPR spectroscopic assays. Source data can be found in *Figure 5—source data 2*.

## End-point assays in $D_2O$

Enzyme was activated as described in 'End-point activity assays with HypD variants.' Assays contained 0.8 mM NADH, 3 μM P5CR, 0.2 mM Hyp, and 0.3 μM HypD (HypD added as a mixture of activation components) in Tris buffer made with $D_2O$ (20 mM Tris-HCl pH 7.5, 100 mM KCl). Assays were carried out in triplicate and were initiated by adding Hyp into reaction mixtures, which were then incubated for 21 hr at 22°C. Upon removal from the anaerobic chamber, reactions were quenched with a 2 × volume of methanol, and precipitated protein was removed by centrifugation (15,200 g, 10 min). LC-MS/MS analysis of Pro, Hyp, and deuterated products were performed on an Agilent 6410 Triple Quadrupole LC-MS instrument (Agilent Technologies) using a Luna SCX column (5 μm, 100 Å, 50 × 2.0 mm, Phenomenex). Precursor and product ions listed in *Figure 6—source data 1* were monitored.

## Preparative bioconversion of [2,5,5-D3] *trans*-4-hydroxy-L-proline and detailed characterization of [2,4,5-D3] L-proline

All enzymes (HypD, HypD-AE, and P5CR) were purified as described above. Both the activation reaction and preparative bioconversion of [2,5,5-D3] *trans*-4-hydroxy-L-proline (2,5,5-D3-Hyp) were

conducted in an MBraun chamber. For the activation reaction, solutions of acriflavine (100 µM final conc.) and bicine (50 mM final conc.) were added to reaction buffer (20 mM Tris pH 7.5, 100 mM KCl) and mixed. HypD-AE (60 µM final conc.) was slowly added to the solution and incubated about 10 inches from the Mbraun chamber light for 30 min. HypD (15 µM final conc.) and AdoMet (1.5 mM final conc.) were added to the activation reaction to a final reaction volume of 7.46 mL (all amounts of activation components were calculated using this final volume). The activation reaction was incubated in a 15 mL falcon tube about 10 inches from the Mbraun chamber light for 2 hr.

For the preparative bioconversion, NADH (4 equiv., 0.8 mM final conc.), P5CR (0.015 equiv., 3 µM final conc.), and activation reaction (HypD – 0.0015 equiv., 0.3 µM final conc.) were diluted with reaction buffer (20 mM Tris pH 7.5, 100 mM KCl) to a final reaction volume of 373 mL in a 500 mL glass bottle. The reaction was initiated by adding 2,5,5-D3-Hyp (1 equiv., 0.2 mM final conc., 10 mg total), mixed, and incubated at 22°C without agitation. Conversion to product was monitored by LC-MS/MS using the protocol outlined above in 'End point assays in $D_2O$' until no substrate could be detected. Upon full conversion, the reaction was removed from the MBraun chamber and enzymes were precipitated with aqueous HCl (1 M, until pH ~1–2). Precipitated protein was removed by centrifugation (28,000 g for 15 min.) and subsequent filtration of supernatant through a 0.22 µm filter. Filtrate was purified by strong cation exchange chromatography. Dowex resin was slurry-packed with methanol and washed with ~100 mL deionized water. The resin was acidified with HCl (1 M) until the flow through was pH <2. The resin was neutralized with deionized water and acidic filtrate was added to the resin. The resin was washed with ~100 mL deionized water, and product was eluted with $NH_4OH$ (1 M). Product-containing fractions, as determined by LC-MS, were lyophilized to dryness. $^1H$ NMR, $^{13}C$ NMR, COSY, and $^1H$-$^{13}C$ HSQC spectra were collected and compared to published $^1H$ NMR and $^{13}C$ NMR spectra for L-proline to assign peaks (Spectral Database for Organic Compounds). An authentic standard of L-proline was used to obtain an $^1H$-$^{13}C$ HSQC spectrum.

2,4,5-D3 L-proline: $^1H$ NMR (400 MHz, $D_2O$) δ 3.18 (m, 1H, $C^5$–H), 2.24 (dd, $J$ = 12.7, 7.1 Hz, 1H, $C^3$–H), 2.00–1.84 (m, 2H, $C^3$–H and $C^4$–H). $^{13}C$ NMR (101 MHz, $D_2O$) δ 175.16, 60.92 (t), 45.70 (t), 28.87, 23.43 (t). HRMS (ESI-TOF) calculated for $C_5H_7D_3NO_2$ [M + H]$^+$: 119.0894, found: 119.0899.

## Acknowledgements

This work was supported in part by National Institutes of Health (NIH) Grant Nos. R01 GM069857 (CLD), R35 GM126982 (CLD), F32 GM129882 (MCA), and R56 AR044276 (RTR); the National Science Foundation (NSF) Graduate Research Fellowship under Grant Nos. 1122374 (LRFB); Harvard University (EPB); a Packard Fellowship for Science and Engineering (2013–39267) (EPB); the NSERC Postgraduate Scholarship-Doctoral Program (YYH); and an Arnold O Beckman Postdoctoral Fellowship (BG). CLD is a Howard Hughes Medical Institute (HHMI) Investigator. LRFB is a recipient of a Dow Fellowship at MIT and a Gilliam Fellowship from HHMI. We would like to thank Michael A Funk for purifying wild-type HypD for protein crystallography experiments and for his help in solving the initial structure of HypD. We would also like to thank Talya S Levitz and Rohan Jonnalagadda for help with LC-MS acquisition. This work is based upon research conducted at the Northeastern Collaborative Access Team beamlines, which are funded by the National Institute of General Medical Sciences from the National Institutes of Health (P41 GM103403). This research used resources of the Advanced Photon Source, a U.S. Department of Energy (DOE) Office of Science User Facility operated for the DOE Office of Science by Argonne National Laboratory under Contract No. DE-AC02-06CH11357. This work was completed in part with resources at the MIT Center for Environmental Health Sciences (P30-ES002109) and the MIT Department of Chemistry Instrumentation Facility with the help of Walt Massefski and John Grimes.

## Additional information

### Funding

| Funder | Grant reference number | Author |
| --- | --- | --- |
| National Institutes of Health | R01 GM069857 | Catherine L Drennan |

| | | |
|---|---|---|
| National Institutes of Health | R35 GM126982 | Catherine L Drennan |
| National Institutes of Health | R56 AR044276 | Ronald T Raines |
| National Science Foundation | Graduate Research Fellowship Grant No. 1122374 | Lindsey RF Backman |
| David and Lucile Packard Foundation | Packard Fellowship for Science and Engineering (2013-39267) | Emily P Balskus |
| Natural Sciences and Engineering Research Council of Canada | Postgraduate Scholarship-Doctoral Program | Yolanda Y Huang |
| Arnold and Mabel Beckman Foundation | Postdoctoral Fellowship | Brian Gold |
| Howard Hughes Medical Institute | HHMI Investigator | Catherine L Drennan |
| Howard Hughes Medical Institute | Gilliam Graduate Fellowship | Lindsey RF Backman |
| Dow Chemical Company | Dow Graduate Fellowship at MIT | Lindsey RF Backman |
| National Institutes of Health | F32 GM129882 | Mary C Andorfer |

The funders had no role in study design, data collection and interpretation, or the decision to submit the work for publication.

## Author contributions
Lindsey RF Backman, Yolanda Y Huang, Conceptualization, Data curation, Formal analysis, Investigation, Visualization, Writing - original draft, Writing - review and editing; Mary C Andorfer, Data curation, Formal analysis, Investigation, Visualization, Writing - review and editing; Brian Gold, Formal analysis, Investigation, Visualization, Writing - review and editing; Ronald T Raines, Supervision, Validation, Writing - review and editing; Emily P Balskus, Catherine L Drennan, Conceptualization, Resources, Formal analysis, Supervision, Funding acquisition, Validation, Investigation, Visualization, Writing - original draft, Project administration, Writing - review and editing

## Author ORCIDs
Lindsey RF Backman ⬤ https://orcid.org/0000-0002-0323-1336
Yolanda Y Huang ⬤ http://orcid.org/0000-0003-1263-1515
Mary C Andorfer ⬤ https://orcid.org/0000-0003-3406-2341
Brian Gold ⬤ https://orcid.org/0000-0002-3534-1329
Ronald T Raines ⬤ https://orcid.org/0000-0001-7164-1719
Emily P Balskus ⬤ https://orcid.org/0000-0001-5985-5714
Catherine L Drennan ⬤ https://orcid.org/0000-0001-5486-2755

## Decision letter and Author response
Decision letter https://doi.org/10.7554/eLife.51420.sa1
Author response https://doi.org/10.7554/eLife.51420.sa2

# Additional files

## Supplementary files
• Transparent reporting form

## Data availability
Diffraction data validation reports have been uploaded to Protein Data Bank under 6VXC and 6VXE.

The following datasets were generated:

| Author(s) | Year | Dataset title | Dataset URL | Database and Identifier |
|---|---|---|---|---|
| Backman LRF, Drennan CL | 2020 | Crystal structure of hydroxyproline dehydratase (HypD) from Clostridioides difficile | https://www.rcsb.org/structure/6VXC | RCSB Protein Data Bank, 6VXC |
| Backman LRF, Drennan CL | 2020 | Crystal structure of hydroxyproline dehydratase (HypD) from Clostridioides difficile with substrate trans-4-hydroxy-L-proline bound | https://www.rcsb.org/structure/6VXE | RCSB Protein Data Bank, 6VXE |

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
