## [Decision Letter]

**Acceptance summary:**

The mechanistic and structural work presented in your manuscript provides valuable insights into catalysis performed by the glycyl radical enzyme HypD. In addition to uncovering the elegant mechanistic logic of HypD-catalyzed 4-hydroxyproline dehydration, by describing the pathway by which prolyl hydroxylation is reversed, this work expands our understanding of metabolic processes performed by the gut microbiome.

**Decision letter after peer review:**

Thank you for submitting your article "Molecular basis for catabolism of the abundant metabolite trans-4-hydroxy-L-proline by a microbial glycyl radical enzyme" for consideration by *eLife*. Your article has been reviewed by three peer reviewers, one of whom served as a guest Reviewing Editor, and the evaluation has been overseen by Wendy Garrett as the Senior Editor. The following individual involved in review of your submission has agreed to reveal their identity: Olivier Berteau (Reviewer #2).

The reviewers have discussed the reviews with one another and the Reviewing Editor has drafted this decision to help you prepare a revised submission. While the reviewers agree that HypD is a mechanistically fascinating enzyme of interest, they have raised several important concerns about mechanistic conclusions drawn.

Essential revisions:

1) Further biochemical support of the proposed mechanisms is necessary. Particularly, the position of H-atom abstraction and re-abstraction requires biochemical support. This is especially important as small differences in distances can easily be overcome by enzyme dynamics and change in substrate puckering. Tracking isotope through the reaction is the most straightforward experiment to address this especially since deuterated substrate is commercially available. Performing the reaction in D2O would further strengthen the mechanistic hypothesis put forward.

2) Clarify comparisons between the selected GREs/rewrite as per comment #1 of the reviewer 3 and enhance structural comparison analysis to support (or revise claims of) distinct orientation of the substrate relative to the CXE motif.

3) An alternative mechanism of elimination, as outlined in comment #5 of Rev 3 is conceivable. Authors should address this mechanistic possibility as an alternative or provide rationale against it.

4) Parts of the Discussion are highly speculative. In particular, the discussion about the stabilization of the putative C5 radical through interaction with the antibonding orbital of the hydroxyl group is vague, and its relevance to catalysis unclear. These highly speculative aspects of the Discussion should be substantially shortened and clearly indicated as a plausible rationale, or removed.

5) Similarly, in another speculative aspect of the Discussion, the authors are proposing that deprotonation of the enamine radical cation intermediate is required for the H-atom abstraction from Cys thiol, despite the low BDE of the S-H bond. Analysis of BDEs and the thermodynamic landscape of the reaction would shed light on whether deprotonation is truly required.

Additionally, the reviewers have identified to several minor revisions, as indicated below.

Reviewer #1:

Glycyl radical enzyme HypD, abundant in both gut microbiome and in prominent bacterial pathogens, catalyzes dehydration of Hyp to P5C. The key structural features include Gly loop characteristic of glycyl radical enzymes, and the Cys loop that harbors catalytic Cys, positioned in the vicinity of each other to enable formation of the thiyl radical. Moreover, unique hydrogen bonding interactions with Glu of CXE motif support protonation of the hydroxyl moiety of the substrate, facilitating its elimination. Hydrophobic packing and aromatic interactions with substrate proline further enable binding and catalysis in this unique GER pocket. Catalytic roles of Gly and Cys are further supported by mutagenesis, as are functions of residues that interact with the substrate. Authors have further performed EPR analysis to monitor the ability of mutants to form the glycyl radical, as well as careful quantitation of Pro (reduced P5C) to evaluate kinetic parameters of the mutants. Together, the findings support the mechanism proposed. Among notable features are: 1. the a-aminoalkyl radical intermediate which differs in its stabilization from the commonly formed a-amino alkyl radicals stabilized by the captodative effects; 2. binding orientations that supports elimination of the hydroxyl group and 3. Deprotonation of the enamine radical cation to generate a-iminyl radical.

Summary of substantive concerns: The manuscript would be significantly improved if the following questions are addressed:

1) Positioning of Hyp in active site suggest H atom abstraction, however this is not clear from the perspective under which structures are shown in Figures 2B and C, where it appears that C4 H atom is closer to the thiyl radial. What is the distance to C4, and how does it change with different puckering of the ring?

2) The mechanism proposed in Figure 6 would strongly benefit from further experimental support. Have authors been able to track the transfer of deuterium from C5 to C4 carbon? This experiment would be a strong addition to the structure of enzyme-bound substrate and mutagenesis presented in support of the mechanisms, and would serve as an important test to the mechanistic proposal put forward.

Reviewer #2:

In this manuscript, Backman and co-workers report the structure and functional investigation of HypD, a newly discovered glycyl radical enzyme (GRE) involved in the metabolism of trans-4-hydroxy-L-proline (Hyp), a major metabolite in the human GI tract. The authors succeeded to solve the high-resolution structure of HypD alone and in the presence of its substrate, allowing them to identify key active-site residues. Based on these structures, mutagenesis analysis, spectroscopic and biochemical data, the authors propose a unique mechanism for the conversion of Hyp to (S)-Δ1-pyrroline-5-carboxylate, implying notably a putative aminoalkyl radical intermediate.

The work has been very carefully executed and most hypotheses have been experimentally validated. The elimination of the hydroxyl group of Hyp and regeneration of the thiyl radical are the most difficult steps to investigate but the authors have provided solid explanations to support their conclusions.

This study represents a major contribution to our understanding of the human microbiome and enlightens how an important human pathogen (*C. difficile*) metabolizes Hyp and can strive in this complex ecosystem.

Reviewer #3:

The manuscript describes a structural characterization of HypD, a 4-hydroxyproline dehydratase. HypD is a member of the glycyl radical eliminase family and is responsible for the metabolism of Hyp in gut microbes. The authors solved the structure of HypD in complex with its substrate Hyp. The catalytic residues were verified by site-directed mutagenesis and activity assays. The studies provide the structural basis of the mechanism of HypD catalysis. Therefore, I think the manuscript attracts the interests of the readership of *eLife*. However, although authors make a significant amount of mechanistic conclusions, some of them are not reasonable or relevant. Also, the position of H-atom abstraction and re-abstraction requires biochemical support. Thus, I suggest acceptance of the manuscript after addressing the following concerns:

1) Introduction, fifth paragraph. These sentences do not make chemical sense, and I do not think the way authors are comparing HypD and RNR is appropriate. RNR does not catalyze dehydration, so it is not clear what comparison the first sentence is trying to make. The authors should be careful about the distinction between dehydration and hydroxide elimination. The second sentence is confusing because neither enzyme catalyzes oxidation. Dehydration reaction catalyzed by HypD is not oxidation as the overall oxidation state of the molecule is unchanged before and after the reaction. So, overall, I do not think the way authors are comparing RNR and HypD here is reasonable or informative. I do, however, noticed potential similarity/distinction in the mechanisms of radical assisted hydroxide elimination catalyzed by the two enzymes as well as viperin and other GRE eliminases. So, authors may want to rewrite this section by focusing on such a mechanism.

2) Subsection “HypD active site reveals interesting variations on a GRE eliminase theme”, end of first paragraph. Although the authors state that the orientation of Hyp with respect to C and E residues are distinct from those in other eliminases, the distinction is not clear. Earlier in the paragraph, authors focused on the conservation of CXE motif and the interaction of E with a hydroxyl group, but a distinct catalytic consequence. No structural comparison was made, and thus it is not clear how distinct the "orientation of Hyp with respect to C and E residues" is between HypD and other eliminases.

3) The authors made strong conclusions about the sites of H-atom abstraction and re-abstraction based on the difference of 0.5-0.6 A between two possible sites, which could easily be perturbed by a dynamic motion of the enzyme during catalysis or in the presence of glycyl radical. Thus, biochemical support is needed. Has an isotope experiment performed on this enzyme? According to the mechanism in Figure 6, the authors should observe a migration of D5 in Hyp to 4-position. 4-[2,5,5-D3]hydroxyproline is available from CIL, and thus it is relatively straightforward to test this hypothesis. Alternatively, if authors perform the reaction in D2O, they may be able to observe wash in of D.

4) –Discussion, third paragraph. I found the discussion about the stabilization of the putative C5 radical through interaction with the antibonding orbital of the hydroxyl group to be vague and speculative. It is not clear such effect is present or critical for the catalysis. Is this effect critical for the selective abstraction of pro-R H5? Or is it essential for catalytic activity at all? If authors want to propose such a mechanism, they should discuss the thermodynamics of such effects on the HypD catalysis.

5) The mechanism of hydroxide elimination in Figure 6 proceeds through aminyl radical carbanion intermediate, which is distinct from the concept of the spin-center shift in Wessig and Muehling, 2007. In the presence of a general base catalyst, the radical assisted β elimination proceeds in a concerted manner. A similar mechanism is conceivable for HypD, in which C5 radical intermediate undergoes concerted N-deprotonation, radical migration from C5 to C4 and elimination of 4-OH. How did the authors eliminate such a mechanism?

6) –Discussion, sixth paragraph. The authors are proposing that deprotonation of an examine radical cation intermediate is required for the H-atom abstraction from thiol. However, BDE of alkyl thiol SH bonds are also relatively low (86-88 kcal/mol; Handbook of BDE in organic compounds). Thus, the H-atom abstraction from Cys is unlikely a challenging step. I would suggest the authors compare these BDE's with that of the C5-H bond and discuss the thermodynamic landscape of the reaction.

---

## [Author Response]

Essential revisions:1) Further biochemical support of the proposed mechanisms is necessary. Particularly, the position of H-atom abstraction and re-abstraction requires biochemical support. This is especially important as small differences in distances can easily be overcome by enzyme dynamics and change in substrate puckering. Tracking isotope through the reaction is the most straightforward experiment to address this especially since deuterated substrate is commercially available. Performing the reaction in D2O would further strengthen the mechanistic hypothesis put forward.

We have carried out the suggested biochemical experiments. We find that when the HypD assay is performed with 2,5,5-D3-Hyp, a product of 2,4,5-D3-L-Pro is formed, suggesting that a D atom is ultimately transferred from C5 to C4, consistent with our proposed mechanism.

2) Clarify comparisons between the selected GREs/rewrite as per comment #1 of the reviewer 3 and enhance structural comparison analysis to support (or revise claims of) distinct orientation of the substrate relative to the CXE motif.

We have rewritten this section of the paper.

3) An alternative mechanism of elimination, as outlined in comment #5 of Rev 3 is conceivable. Authors should address this mechanistic possibility as an alternative or provide rationale against it.

As reviewer 3 indicated, our data cannot rule out the possibility that deprotonation of the proposed aminyl radical occurs simultaneously with elimination of the C4 OH group. We have now mentioned this possibility in the Discussion.

4) Parts of the Discussion are highly speculative. In particular, the discussion about the stabilization of the putative C5 radical through interaction with the antibonding orbital of the hydroxyl group is vague, and its relevance to catalysis unclear. These highly speculative aspects of the Discussion should be substantially shortened and clearly indicated as a plausible rationale, or removed.

We have removed this proposal from the main text.

5) Similarly, in another speculative aspect of the Discussion, the authors are proposing that deprotonation of the enamine radical cation intermediate is required for the H-atom abstraction from Cys thiol, despite the low BDE of the S-H bond. Analysis of BDEs and the thermodynamic landscape of the reaction would shed light on whether deprotonation is truly required.

This section was poorly worded. We have rewritten it.

Additionally, the reviewers have identified to several minor revisions, as indicated below.Reviewer #1:[…] Summary of substantive concerns: The manuscript would be significantly improved if the following questions are addressed:1) Positioning of Hyp in active site suggest H atom abstraction, however this is not clear from the perspective under which structures are shown in Figures 2B and C, where it appears that C4 H atom is closer to the thiyl radial. What is the distance to C4, and how does it change with different puckering of the ring?

Distances between hydrogens on Hyp and Cys434 are shown in Figure 8. We now point readers toward Figure 8 in the legend of Figure 2.

2) The mechanism proposed in Figure 6 would strongly benefit from further experimental support. Have authors been able to track the transfer of deuterium from C5 to C4 carbon? This experiment would be a strong addition to the structure of enzyme-bound substrate and mutagenesis presented in support of the mechanisms, and would serve as an important test to the mechanistic proposal put forward.

We have done these experiments and they are now included in the manuscript (also see response 1 above).

Reviewer #3:The manuscript describes a structural characterization of HypD, a 4-hydroxyproline dehydratase. HypD is a member of the glycyl radical eliminase family and is responsible for the metabolism of Hyp in gut microbes. The authors solved the structure of HypD in complex with its substrate Hyp. The catalytic residues were verified by site-directed mutagenesis and activity assays. The studies provide the structural basis of the mechanism of HypD catalysis. Therefore, I think the manuscript attracts the interests of the readership of eLife. However, although authors make a significant amount of mechanistic conclusions, some of them are not reasonable or relevant. Also, the position of H-atom abstraction and re-abstraction requires biochemical support. Thus, I suggest acceptance of the manuscript after addressing the following concerns:1) Introduction, fifth paragraph. These sentences do not make chemical sense, and I do not think the way authors are comparing HypD and RNR is appropriate. RNR does not catalyze dehydration, so it is not clear what comparison the first sentence is trying to make. The authors should be careful about the distinction between dehydration and hydroxide elimination. The second sentence is confusing because neither enzyme catalyzes oxidation. Dehydration reaction catalyzed by HypD is not oxidation as the overall oxidation state of the molecule is unchanged before and after the reaction. So, overall, I do not think the way authors are comparing RNR and HypD here is reasonable or informative. I do, however, noticed potential similarity/distinction in the mechanisms of radical assisted hydroxide elimination catalyzed by the two enzymes as well as viperin and other GRE eliminases. So, authors may want to rewrite this section by focusing on such a mechanism.

The point was that the substrates of HypD and RNR are similar. We have rephrased to make this point more clearly and have shortened this section.

2) Subsection “HypD active site reveals interesting variations on a GRE eliminase theme”, end of first paragraph. Although the authors state that the orientation of Hyp with respect to C and E residues are distinct from those in other eliminases, the distinction is not clear. Earlier in the paragraph, authors focused on the conservation of CXE motif and the interaction of E with a hydroxyl group, but a distinct catalytic consequence. No structural comparison was made, and thus it is not clear how distinct the "orientation of Hyp with respect to C and E residues" is between HypD and other eliminases.

We have rephrased this section of the paper.

3) The authors made strong conclusions about the sites of H-atom abstraction and re-abstraction based on the difference of 0.5-0.6 A between two possible sites, which could easily be perturbed by a dynamic motion of the enzyme during catalysis or in the presence of glycyl radical. Thus, biochemical support is needed. Has an isotope experiment performed on this enzyme? According to the mechanism in Figure 6, the authors should observe a migration of D5 in Hyp to 4-position. 4-[2,5,5-D3]hydroxyproline is available from CIL, and thus it is relatively straightforward to test this hypothesis. Alternatively, if authors perform the reaction in D2O, they may be able to observe wash in of D.

We performed these experiments. See response to overall comment #1 above.

4) –Discussion, third paragraph. I found the discussion about the stabilization of the putative C5 radical through interaction with the antibonding orbital of the hydroxyl group to be vague and speculative. It is not clear such effect is present or critical for the catalysis. Is this effect critical for the selective abstraction of pro-R H5? Or is it essential for catalytic activity at all? If authors want to propose such a mechanism, they should discuss the thermodynamics of such effects on the HypD catalysis.

We have removed this proposal from the main text.

5) The mechanism of hydroxide elimination in Figure 6 proceeds through aminyl radical carbanion intermediate, which is distinct from the concept of the spin-center shift in Wessig and Muehling, 2007. In the presence of a general base catalyst, the radical assisted β elimination proceeds in a concerted manner. A similar mechanism is conceivable for HypD, in which C5 radical intermediate undergoes concerted N-deprotonation, radical migration from C5 to C4 and elimination of 4-OH. How did the authors eliminate such a mechanism?

As the reviewer indicates, our data cannot rule out the possibility that deprotonation of the proposed aminyl radical occurs simultaneously with elimination of the C4 OH group. We have now mentioned this possibility in the Discussion.

6) –Discussion, sixth paragraph. The authors are proposing that deprotonation of an examine radical cation intermediate is required for the H-atom abstraction from thiol. However, BDE of alkyl thiol SH bonds are also relatively low (86-88 kcal/mol; Handbook of BDE in organic compounds). Thus, the H-atom abstraction from Cys is unlikely a challenging step. I would suggest the authors compare these BDE's with that of the C5-H bond and discuss the thermodynamic landscape of the reaction.

We have rewritten this section of the Discussion and have shortened this part of the Discussion as the overall consensus of the reviewers was that the Discussion was too long.